# Tumor-derived NKG2D ligand sMIC reprograms NK cells to an inflammatory phenotype through CBM signalosome activation

Payal Dhar [1], Fahmin Basher[2], Zhe Ji [3,4], Lei Huang[5], Si Qin[1], Derek A. Wainwright [6,7], Jerid Robinson[8], Shaye Hagler[8], Jing Zhou[8], Sean MacKay[8] & Jennifer D. Wu [1,7 ✉]

Natural Killer (NK) cell dysfunction is associated with poorer clinical outcome in cancer patients. What regulates NK cell dysfunction in tumor microenvironment is not well understood. Here, we demonstrate that the human tumor-derived NKG2D ligand soluble MIC (sMIC) reprograms NK cell to secrete pro-tumorigenic cytokines with diminished cytotoxicity and polyfunctional potential. Antibody clearing sMIC restores NK cell to a normal cytotoxic effector functional state. We discovered that sMIC selectively activates the CBM-signalosome inflammatory pathways in NK cells. Conversely, tumor cell membrane-bound MIC (mMIC) stimulates NK cell cytotoxicity through activating PLC2γ2/SLP-76/Vav1 pathway. Ultimately, antibody targeting sMIC effectuated the in vivo anti-tumor effect of adoptively transferred NK cells. Our findings uncover an unrecognized mechanism that could instruct NK cell to a dysfunctional state in response to cues in the tumor microenvironment. Our findings provide a rationale for co-targeting sMIC to enhance the efficacy of the ongoing NK cell-based cancer immunotherapy.

[1] Department of Urology, Feinberg School of Medicine, Northwestern University, Chicago, IL, USA. [2] Division of General Internal Medicine, Department of Medicine, University of Miami, Miami, FL, USA. [3] Department of Pharmacology, Feinberg School of Medicine, Northwestern University, Chicago, IL, USA. [4] Department of Biochemistry and Molecular Genetics, Feinberg School of Medicine, Northwestern University, Chicago, IL, USA. [5] Center for Research Informatics, The University of Chicago, Chicago, IL, USA. [6] Department of Neurological Surgery, Feinberg School of Medicine, Northwestern University, Chicago, IL, USA. [7] Department of Microbiology and Immunology, Feinberg School of Medicine, Northwestern University, Chicago, IL, USA. [8] Isoplexis Corporation, Branford, CT, USA. ✉email: jennifer.wu@northwestern.edu

Natural killer (NK) cells are a subset of innate lymphocytes that execute their immune effect predominantly through mediating cytotoxic function. NK cells are at the forefront of development for cancer immunotherapy[1]. Canonically, the activation of NK cells is instructed by engagement of its surface activating and inhibitory receptor repertoire, independent of classical MHC I antigen presentation[1]. Natural killer group 2, member D (NKG2D) is a major activating receptor expressed by all NK cells[2,3]. Engaging NKG2D by forced NKG2D-L expression on the surface of target cells is sufficient to activate NK cell cytotoxicity and to control tumor growth in experimental animal models[4–7]. NKG2D-deficient mice have defective tumor immune surveillance against both solid and hematological tumors[8]. These evidence signify the magnitude of activating the NKG2D/ NKG2D-L axis in mediating NK cell cytotoxicity against tumors. However, NKG2D ligands are abundantly found in advanced human tumors[9,10]; in some cases, the high abundancy of NKG2D ligands in tumors associates with poorer clinical outcome[11–15]. The underlying mechanisms of such a dichotomy are not well-elucidated.

NKG2D recognizes diverse families of ligands that in humans are known to be induced by oncogenic insults, viral infection, or inflammatory stress[2,16]. These ligands are generally not expressed by normal cells. Human NKG2D ligands include the MHC I chain-related family of molecules MICA and MICB (collectively referred as MIC) and the UL-16-binding family of molecules ULBP1-6[2,16]. Mice do not express orthologs of MIC[17,18], but express mouse MHC I family molecules of H60 and Rae-1 and mouse ULBP-like molecules of MULT[2,16]. The co-existence of diverse families of NKG2D ligands was thought to be attributed to co-evolution or gene duplication during pathogen infections[19]. Among the identified human NKG2D ligands, the MIC family molecules are known to be the most commonly expressed in association with oncogenic insults[16]. Induced MIC expression on tumor cell surface presumably provides active tumor immune surveillance via activation of NK cells and co-stimulates other immune effector cells to curtail disease progression[20]. Shedding MIC is considered a major strategy for tumors to evade NKG2D-mediated immune surveillance[21,22]. Inhibition or clearance of tumor cell shedding of human MIC preserves and potentiates NK cell-driven tumor immunity in experimental models[23,24], heightening the magnitude of MIC/NKG2D axis in directing NK cell cytotoxic function to control tumors. However, it remains largely controversial over the impact of NKG2D-L on NK cell function, with existing conflicting data demonstrating that NKG2D-L could stimulate or suppress NK cell anti-tumor function[4,5,14,15,24–27]. The mechanisms underlying these conflicting data were not well-defined. These confusing findings limited the viability of harnessing the NKG2D/NKG2D-L pathway to improve NK cell-based immunotherapy of cancer. Clearly, there is a need to discern the mechanisms of how the human NKG2D ligand MIC regulates NK cell function in the context of the tumor.

The well-documented co-existence of soluble and membrane NKG2D ligands in human tumors may give rise to the complexity and thus the conflicting conclusions of current data. In this study, we aim to delineate the impact of sMIC and cell surface-bound membrane MIC (mMIC) on NK cell function with the understanding at molecular levels. We present multiple lines of evidence that tumor-derived sMIC skews NK cell to a dysfunctional pro-inflammatory pro-tumorigenic phenotype with diminished cytotoxic effector function and polyfunctionality, whereas mMIC stimulates NK cell cytotoxic function. We demonstrate that NKG2D engagement of sMIC selectively activates CARMA1-BCL10-MALT1 (CBM) signalosome and skews NK cell to secrete pro-tumorigenic cytokines. Conversely, NKG2D engagement of mMIC activates PLCγ2/Vav1/SLP-76 and ERK/JNK pathway,

thereby enabling NK cell cytotoxic effector function. For the therapeutic proof-of-concept, we demonstrate that administration of an sMIC-clearing nonblocking antibody effectuated NK cell in vivo activity in controlling MIC+ tumors. Our findings uncover a previously undescribed mechanism that askews NK cell to a malfunctioning status. We also differentiate the impact and underlying mechanisms whereby sMIC and mMIC dictate NK cell function. Our findings provide the rationale for targeting sMIC to enhance NK cell-based therapy of cancer[28].

## Results

**sMIC stimulation reprograms NK cells to a polarized pro-inflammatory gene transcriptome.** How human tumor-derived sMIC regulates NK cell function is not fully understood, although it has been shown that sMIC can downregulate NK cell surface NKG2D expression and cytotoxic function[22,24,29]. To elucidate the full impact of sMIC on NK cell function, we performed RNA-sequencing analyses on primary human and mouse NK cells that were stimulated with recombinant sMIC alone and sMIC in presence of the nonblocking sMIC-clearing mouse monoclonal antibody (mAb) B10G5, as previously described[30]. Differential gene expression analysis revealed a global effect of sMIC on NK cell transcriptional reprogramming, with 4950 genes differentially expressed in sMIC-stimulated versus unstimulated human NK cells (Fig. 1a and Supplementary Fig. 1a) and 6503 genes differentially expressed in sMIC-stimulated versus unstimulated mouse NK cells (Fig. 1b and Supplementary Fig. 2a). These sMIC-stimulated differentially expressed genes were restored to the expression level in unstimulated NK cells by the addition of the sMIC-clearing antibody B10G5 (Fig. 1a, b). Gene ontology (GO) revealed that, compared to unstimulated NK cells, sMIC-stimulated NK cells were enriched for genes associated with inflammation and pro-tumorigenic cytokines and chemokines, such as *RELB, CCL1, CCL3, NFKB2, CCL4*, and *IL-10* for human NK cells and *CCL5, NFKB2, RELB, CCL4, RELA*, and *TGFB1* for mouse NK cells (Fig. 1c, d). In contrast, genes associated with NK cell cytotoxic effector function, such as *NCR1, CD226, CD160, GZMM, KLRK1 (NKG2D)* for human NK cells and *IFNG, GZMB, CD226, KLRK1 (NKG2D), CD160* for mouse NK cells, were downregulated with sMIC stimulation (Fig. 1c, d). The sMIC-associated modulations of representative gene expression associated with inflammation and cytotoxicity were further validated by qRT-PCR (Fig. 1e and Supplementary Figs. 1b, 2b). Together, these gene expression data suggest that sMIC could profoundly skew NK cells to a polarized function of pro-tumorigenic cytokine and chemokine production beyond downregulating surface NKG2D expression.

**NK cells from aggressive sMIChi tumors displayed a distinct inflammatory expression profile with diminished cytotoxic potential.** To further elucidate the impact of sMIC on NK cell function in a complex tumor microenvironment, we performed single-cell RNA-sequencing (scRNAseq) transcriptomic profiling of NK cells isolated from respective sMIChi and sMIClo prostate tumors of TRAMP/MIC mice (Fig. 2a, b and Supplementary Fig. 3a). Due to the factor that no homolog of human MIC was identified in mice, to understand the human MIC-NKG2D onco-immune biology, we had generated the double transgenic TRAMP/MIC mice in which human native MIC was engineered to be specifically expressed in prostate tumors[6]. Previously, we had demonstrated that TRAMP/MIC mice with high levels of tumor-shed sMIC developed poorly differentiated (PD) tumors, whereas those with low levels of tumor-shed sMIC developed well-differentiated (WD) tumors[6]. Based on the canonical cell surface markers and the K-nearest neighbor (KNN) graph-based

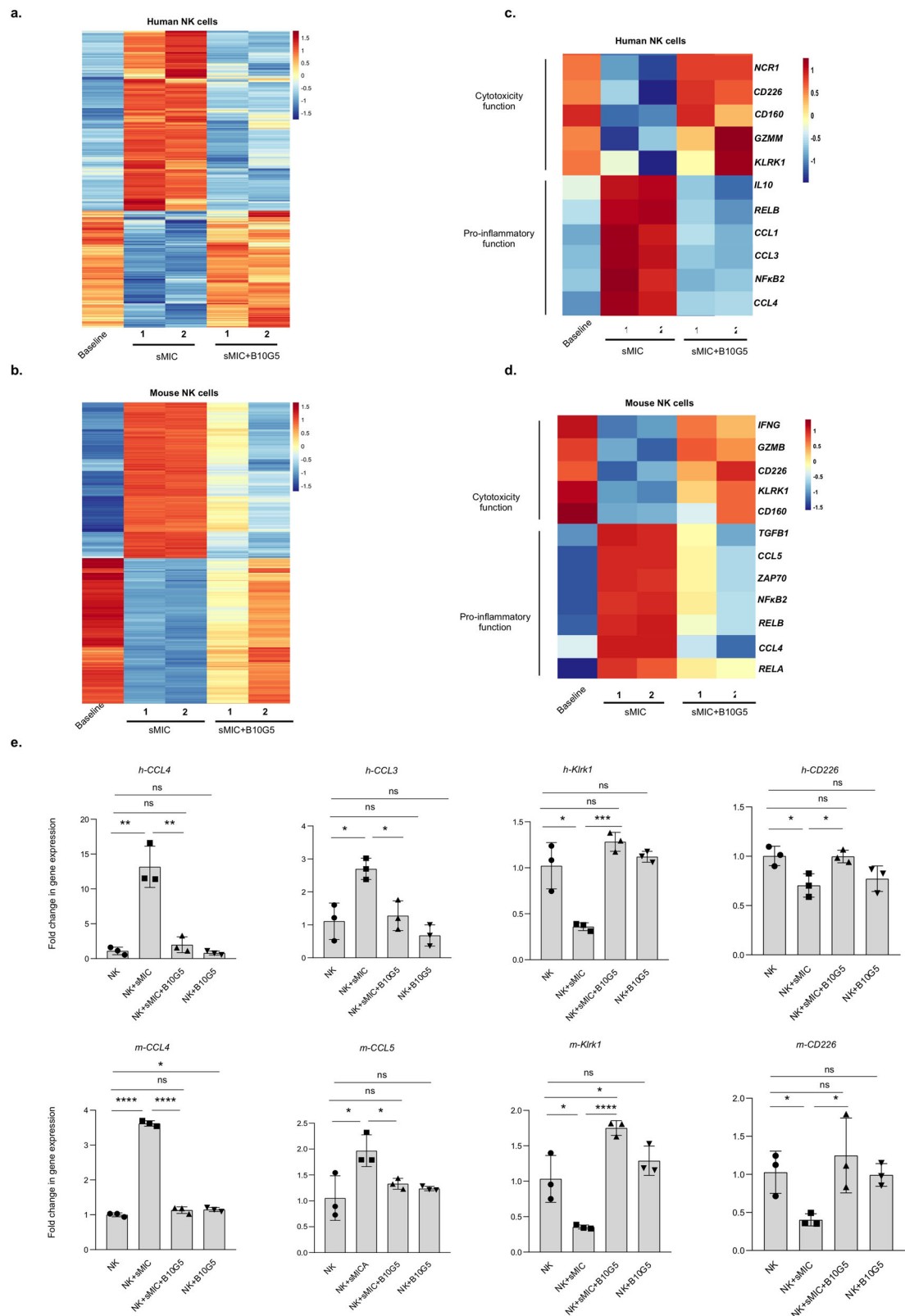

clustering algorithm, a total of 28 distinct clusters were identified in CD45[+] TILs and visualized as a UMAP (Uniform Manifold Approximation and Projection) (Fig. 2c and Supplementary Fig. 3b). Gene set enrichment (GSEA) and gene ontology (GO) pathway analyses of NK cell clusters revealed distinct functional differentiation of NK cells in tumors with sMIC[hi] and tumors with sMIC[lo] (Fig. 2d). Using the HIPPO method[31] with

prototypic function-associated markers, such as Ncr1, Itga1, Klrk1, Gzma, Gzmb, Tnfsf10, Tbx21, and Eomes, two functional subsets of NK cells were identified, the subset with pro-inflammatory function and the subset with cytotoxic function (Fig. 2e and Supplementary Fig. 3c). The two functional subsets of NK cells were differentially enriched in sMIC[lo] (WD) and sMIC[hi] (PD) tumors (Fig. 2e–g). In sMIC[lo] tumors, NK cells had a gene

**Fig. 1 sMIC stimulation reprograms NK cells toward pro-inflammatory gene transcriptome.** Primary human NK cells enriched from PBMCs were stimulated with 100 ng/ml of recombinant sMICA in the absence or presence of sMIC-clearing monoclonal antibody B10G5 for 18 h. Total RNA was isolated for bulk RNA-sequencing analysis. **a** Heatmap representing the overall number of differentially expressed genes in human NK cells stimulated with sMICA, compared to the cells stimulated with sMICA in presence of B10G5. Primary mouse NK cells isolated from spleens of Rag$^{-/-}$ mice were stimulated with 100 ng/ml of recombinant sMICB in the absence or presence of B10G5 for 18 h. Total RNA was isolated for bulk RNA-sequencing analysis. **b** Heatmap representing the overall number of differentially expressed genes in mouse NK cells stimulated with sMICB, compared to the cells stimulated with sMICB in presence of B10G5. **c** Heatmap highlighting the key differentially expressed genes related to cytotoxicity and inflammatory response pathways in human NK cells upon stimulation with sMICA and sMICA+B10G5. **d** Heatmap highlighting the key differentially expressed genes related to cytotoxicity and inflammatory response pathways in mouse NK cells upon stimulation with sMICB and sMICB+B10G5. **e** Validation of representative genes associated with NK cell cytotoxicity and pro-inflammatory function (represented in (**b**) and (**d**)) by qRT-PCR. *$P < 0.05$, **$P < 0.01$, ***$P < 0.001$, and ****$P < 0.0001$ (Student's $t$ test; two tailed).

signature predominantly associated with cytotoxicity. Conversely, NK cells from sMIC$^{hi}$ tumors displayed a gene signature predominantly associated with pro-inflammatory function (e.g., *CCL3, CCL4, CCR2*) and diminished cytotoxicity (Fig. 2e–g). Flow cytometry analyses of tumor-infiltrating NK cells from sMIC$^{lo}$ and sMIC$^{hi}$ tumors validated that NK cells from sMIC$^{hi}$ tumors had diminished cytotoxic effector function and marked reduction in the capacity to produce anti-tumor cytokines, such as IFNγ and TNFα (Fig. 2h and Supplementary Fig. 3d). Noteworthy, consistent with our previous findings[6], the number of NK cells in sMIC$^{hi}$ tumors was significantly reduced as compared to sMIC$^{lo}$ tumors. These data suggest an alternative pro-inflammatory transcriptome with the diminished cytotoxic potential of NK cells in association with high levels of tumor-derived sMIC.

**Soluble, not membrane-bound MIC, induces NK cells to produce tumor-promoting inflammatory cytokines and chemokines.** In MIC$^+$ tumors, whether in cancer patients or in the genetically engineered TRAMP/MIC mice, sMIC and mMIC co-exist. How the two forms of MIC regulate NK cell function remains poorly understood and controversial. To delineate which form or whether both forms of MIC in tumors skews NK cell dysfunction as presented above, we generated mouse and human tumor cell lines that, respectively, express sMIC composed of the ectodomain α1–α3 of the MICA or MICB molecule and the non-sheddable membrane-restricted MIC, mMIC, previously described as MICA.A2 or MICB.A2[6,32,33]. We cultured mouse splenic NK cells in the presence of the mouse prostate tumor TRAMP-C2 (TC2) cells, sMICB-expressing TC2 cells (TC2-sMICB), and mMICB-expressing TC2 cells (TC2-mMICB) for 6 and 36 h, respectively. Stimulation with TC2-sMICB cells, but not with TC2-mMICB cells, consistently resulted in a marked enrichment of inflammatory cytokines and chemokines secretion by NK cells as compared to stimulation with TC2 cells (Fig. 3a, b and Supplementary Fig. 4a). Noteworthy, there were no differences in the expression of other mouse NKG2D ligands Rae-1, MULT1, H60 on the tumor cells lines (Supplementary Fig. 4b). In addition, neither expression of sMICB nor mMICB affected on TC2 tumor cell cytokine or chemokine secretion (Supplementary Fig. 4c). The significantly increased representative chemokine and pro-tumorigenic cytokine production by sMIC-stimulated NK cells as compared to unstimulated or mMIC-stimulated NK cells was confirmed by qRT-PCR analyses (Supplementary Fig. 4d, e). This observation was further validated in primary human NK cells co-cultured with sMICA or mMICA expressing human B lymphoblast C1R cells (MIC-negative). Human NK cells secreted a remarkable amount of inflammatory chemokines and pro-tumorigenic cytokines upon stimulation by C1R-sMICA cells, not by C1R or C1R-mMIC cells (Fig. 3c, d). These results suggest that the stimulation of NK cells with sMIC preferentially activates the inflammatory pathways.

To further confirm that sMIC stimulation polarizes NK cell activation to inflammatory chemokine and pro-tumorigenic cytokine production pathways, we stimulated mouse NK cells with purified recombinant sMICB (rsMICB) in the absence and presence of an sMIC-clearing mAb B10G5. Stimulation of NK cells with purified rsMICB resulted in significantly enhanced expression of *IL-10* and *GM-CSF* (Supplementary Fig. 4f, g). The addition of B10G5 resulted in the restoration of NK cell cytokine production to the normal status.

**Soluble MIC activates CBM-signalosome pathways.** NKG2D signals through the adaptor molecule DAP10 or DAP12 that contains the ITAM motif[34]. Following activation, distinct signaling pathways uncoupling NK cell cytokine secretion from cytotoxicity have been demonstrated through the Fyn-ADAP and the CARMA1/BCL10/MALT1 (CBM) signalosome pathway[35]. To understand how sMIC stimulation activates NK cell pro-inflammatory cytokine and chemokine production, we first evaluated the activation of the Fyn-ADAP-CBM-signalosome pathway in primary human NK cells co-cultured with C1R-sMICA and C1R-mMICA cells, respectively. As shown in Fig. 4a and b, co-culturing with C1R-sMICA cells resulted in a marked increase in the phosphorylation of the kinase Fyn and the downstream molecule ADAP in NK cells as compared to co-culturing with C1R-mMICA cells. Correspondingly, exposure to sMIC induced a significant increase in the levels of CARD11, BCL10, and MALT1 that are key components of the CBM-signalosome complex (Fig. 4c–e). The differences in the activation of these key signaling molecules were abrogated when the NK cells were pre-incubated with NKG2D-specific blocking antibody 1D11, confirming that these signaling events are NKG2D-dependent (Supplementary Fig. 5a). Furthermore, phosphorylation of PKC$_\theta$, a T cell-specific kinase important for CBM assembly and classical NF-κB activation[36], was also significantly elevated in NK cells after stimulation with C1R-sMICA cells as compared to stimulation with C1R-mMICA cells (Fig. 4f).

Activation of the CBM signalosome leads to activation NF-κB and consequently induces NK cell secretion of inflammatory cytokines with reduced cytotoxicity[37,38]. As shown in Fig. 4g, within 1 h of stimulation, phosphorylation of the NF-κB subunit p65 was evidently elevated in NK cells stimulated with C1R-sMICA cells as compared to stimulation with C1R or C1R-mMIC cells. Since C1R-sMICA cells and C1R-mMICA cells express GFP as a selection marker, we confirmed that the NK cells collected from the co-culture did not contain tumor cells by re-probing of the blots with an anti-GFP antibody (Supplementary Fig. 5b, c). In parallel to the co-culture system, NK cells were stimulated with recombinant sMIC and plate-bound MIC for different stimulation times of 10, 30, and 60 min. Phosphorylation of the NF-κB subunit p65 was evidently elevated in NK cells stimulated with recombinant sMIC as compared to stimulation with plate-bound MIC at 60 min of the stimulation time period. Differences in

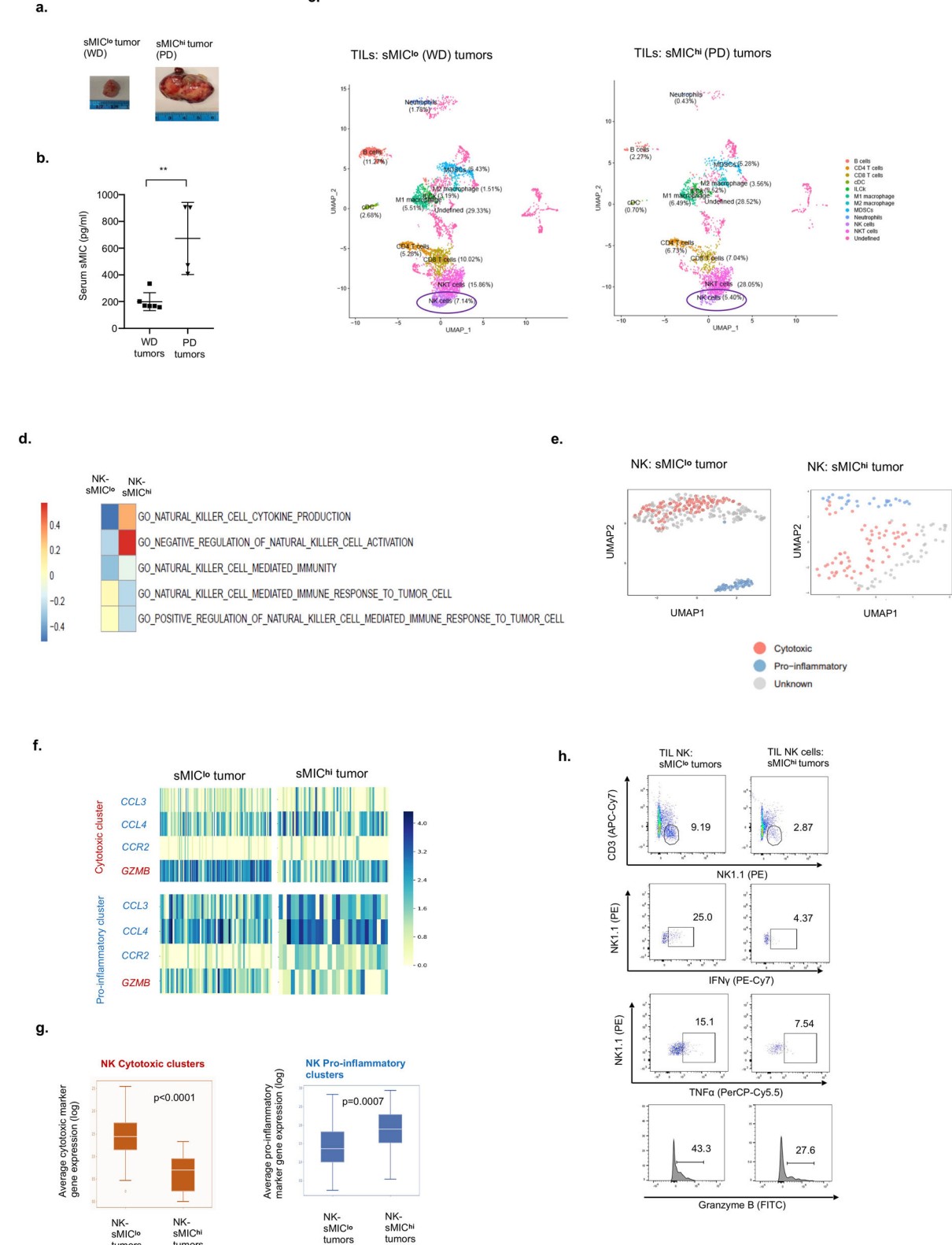

ADAP levels were also evidently higher in recombinant sMIC-stimulated NK cells compared to plate-bound MIC with 10 and 30 min of stimulation (Supplementary Fig. 5d).

We further analyzed the activation of key molecules upstream of the signaling cascade, including the ITAM-phosphorylating Src kinase Lck and products from activation of the secondary intracellular mediator phosphatidylinositol 4,5-bisphosphate (PIP2), diacylglycerol (DAG), and inositol 1,4,5-triphosphate (IP3), in primary human NK cells in response to sMIC and mMIC stimulation (Supplementary Fig. 5e). DAG is known to be the key regulator of PKC-θ activity[39,40]. IP3 is known to regulator NK cell cytotoxic activity through $Ca^{2+}$ mobilization[41]. No apparent differences in Lck activation levels were observed with sMIC versus mMIC stimulation at 10 and 60 min (Supplementary

**Fig. 2 Single-cell expression profiles of tumor-infiltrating NK cells from aggressive sMIC$^+$ tumors display distinct pro-inflammatory functional phenotypes with diminished cytotoxicity.** Tumors from sMICB$^{hi}$ and sMICB$^{lo}$ TRAMP/MIC mice were harvested, and single-cell suspension was obtained using enzymatic digestion. CD45$^+$ tumor-infiltrating lymphocytes were isolated by cell sorting. Single-cell RNA-sequencing analysis of tumor-infiltrating lymphocytes was conducted using the 10× genomics platform. **a** Representative images of (left panel) sMICB$^{lo}$ well-differentiated (WD) and (right panel) sMIC$^{hi}$ poorly differentiated (PD) tumors from TRAMP/MICB mice. **b** Serum levels of sMICB in WD tumors ($n = 6$) and PD tumors ($n = 4$) from TRAMP/MICB mice. **c** UMAP plots illustrating different tumor-infiltrating immune cell populations identified by clustering algorithms in well-differentiated sMICB$^{lo}$ and poorly differentiated sMICB$^{hi}$ tumors. **d** Heatmap representing key gene ontology (GO) pathways affected in NK cells in sMICB$^{lo}$ and sMICB$^{hi}$ tumors identified using gene set enrichment analysis (GSEA). **e** UMAP plots illustrating identified NK cell cytotoxic and pro-inflammatory clusters in the sMICB$^{lo}$ and sMICB$^{hi}$ tumors. **f** Heatmaps representing the gene expression of cytotoxic and pro-inflammatory markers at the single NK cell level in the identified NK cell cytotoxic sub-cluster (top panel) and pro-inflammatory sub-cluster (bottom panel) in the sMICB$^{lo}$ and sMICB$^{hi}$ tumors. **g** Box plots representation of average cytotoxic marker gene expression level (top panel) and average pro-inflammatory marker gene expression level (bottom panel) in NK cells and their comparison between sMICB$^{lo}$ and sMICB$^{hi}$ tumors. **h** Functional analysis of tumor-infiltrating NK cells from sMICB$^{lo}$ and sMICB$^{hi}$ tumors in response to ex vivo PMA/Ionomycin stimulation assessed by flow cytometry. Representative flow cytometry dot plots and histograms demonstrating higher NK1.1$^+$ cell population, higher IFNγ, Granzyme B, and TNFα-producing NK cells from sMICB$^{lo}$ tumors compared to sMICB$^{hi}$ tumors. **$P < 0.01$ (Student's $t$ test; two-tailed).

Fig. 5e), suggesting that Lck activation might be associated with both sMIC and mMIC-mediated signaling events. However, DAG lipase β presented differential dynamics. Upon acute stimulation, e.g., 10 min, DAG lipase β was comparable between sMIC and mMIC stimulation. Upon prolonged stimulation, e.g., 30 min, DAG lipase β was evidently higher with sMIC stimulation (Supplementary Fig. 5e). On the contrary, phosphorylation of IP3 was elevated in NK cells stimulated with mMIC with a short stimulation period (Supplementary Fig. 5e). Given that Ca$^{2+}$ influx reaches its pinnacle within 5 min of NK stimulation, the observation is anticipated[42,43]. Taken together, these data demonstrate that sMIC stimulation activates the CBM-signalosome pathway in NK cells and skews NK cell function towards a pro-inflammatory phenotype.

**Membrane-bound, but not soluble MIC, dictates NK cell cytotoxicity.** To address the effect of mMIC on NK cell cytotoxic function as compared to that of sMIC, we evaluated the cytotoxicity of human NK cell line NKL against C1R-sMICA and C1R-mMICA target cells. As shown in Fig. 5a, NKL cells demonstrated the highest cytotoxicity against C1R-mMICA cells but no or negligible cytotoxicity against C1R-sMIC cells that was comparable to the cytotoxicity against MIC negative control C1R cells. These data suggest that mMIC but not sMIC sensitizes target cells to NK cell cytotoxicity.

Tumors often present with a highly heterogeneous phenotype and a co-existence of tumor cells retaining membrane MIC and some shedding sMIC. We, therefore, assessed whether the presence of sMIC would affect NK cell cytotoxicity against mMIC-expressing target cells. Interestingly, the presence of sMIC inhibited NK cell cytotoxicity against C1R-mMICA cells (Fig. 5b). Importantly, the addition of the nonblocking sMIC-clearing mAb B10G5 restored NK cytotoxic function against C1R-mMICA cells (Fig. 5b). Pre-incubation of NK cells with an NKG2D-specific blocking antibody 1D11 abolished the NK cell cytotoxicity, confirming the NKG2D-dependent cytotoxic effect (Supplementary Fig. 6a). These data clearly demonstrate the opposite effect of mMIC and sMIC on NK cell cytotoxicity against target cells and that only mMIC stimulates NK cell cytotoxic activity. These data also suggest that clearing sMIC with a nonblocking antibody can preserve NK cell cytotoxic function to target tumor cells presenting mMIC.

We further corroborated our findings with mouse NK cells as effector cells against mouse prostate tumor cells TC2 expressing sMICB (TC2-sMICB) and mMICB (TC2-mMICB). NK cells isolated from wild-type mice demonstrated high cytotoxicity against TC2-mMICB, but not TC2-sMICB cells. NKG2D-deficient NK cells did not present significant cytotoxicity against

TC2-mMICB cells (Supplementary Fig. 6b), confirming NKG2D-dependent cytotoxicity against mMIC-expressing target cells. NK cells stimulated with TC2-mMICB showed a significant increase in granzyme B expression by qRT-PCR as compared to the stimulation with TC2-sMICB cells (Supplementary Fig. 6c), further confirming the differential effects of sMIC and mMIC on NK cell cytotoxic function.

Exerting cytotoxic function of NK cells requires the activation of signaling molecules Vav1, PLCγ2, and SLP-76 following receptor activations. The deficiency of these signaling molecules results in the defect in Ca2$^+$ mobilization and degranulation of NK cells[44–46]. Downstream activation of ERK and JNK kinases is similarly crucial for NK cell cytotoxicity since activation of these kinases is required for the microtubule organizing center (MTOC) and cytolytic granule polarization to the NK cell immunological synapse[47–49]. To confirm that sMIC and mMIC differentially regulate NK cell cytotoxic function, we assessed the activation of the PLCγ2/ Vav1/SLP-76 and ERK-JNK kinases by western blotting of human NK cell lysates after co-culture with C1R-sMICA or C1R-mMICA cells. As shown in Fig. 5c–j, co-culture of NK cells with C1R-mMICA cells resulted in a marked increase in the phosphorylation of Vav1, PLCγ2, SLP-76, ERK, and JNK as compared to NK cells co-cultured with C1R-sMICA cells. Increased phosphorylation of ERK and JNK was further confirmed in mouse NK cells when co-cultured with TC2-mMICB cells in comparison to the culture with TC2-sMICB cells (Supplementary Fig. 6d). We further compared stimulation of primary human NK cells with recombinant sMIC in media and with plate-bound immobilized sMIC. With an acute short period of stimulation, a higher level of phosphorylation of ERK and JNK was observed with plate-bound sMIC compared to sMIC, which may due to sMIC immobilization-induced crosslink. However, upon prolonged stimulation, e.g., 60 min, a similarly marked reduction in the phosphorylation of ERK and JNK was seen with sMIC in media and immobilized (Supplementary Fig. 6e). Together, these results demonstrate that cell mMIC stimulation preferentially activates NK cell cytotoxic signaling pathways and thereby enhances their cytotoxic potential.

**Membrane-bound MIC induces higher IFNγ production in NK cells.** NK cells execute their effector functions through cytolytic activity by killing target cells and secretion of IFNγ to regulate adaptive immune responses[50]. To corroborate that membrane-bound MIC has a superior ability to stimulate NK cell effector functions than sMIC, we therefore assessed IFNγ production in the experimental setting of primary human and mouse NK cells, in response to stimulation by sMIC and mMIC at different timepoints. Intriguingly, stimulation with mMIC triggered higher

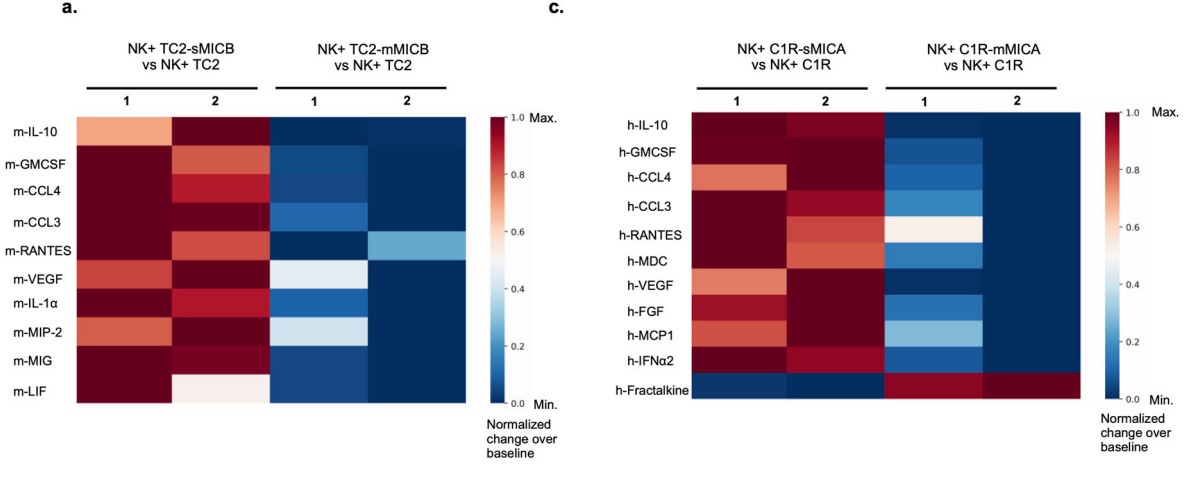

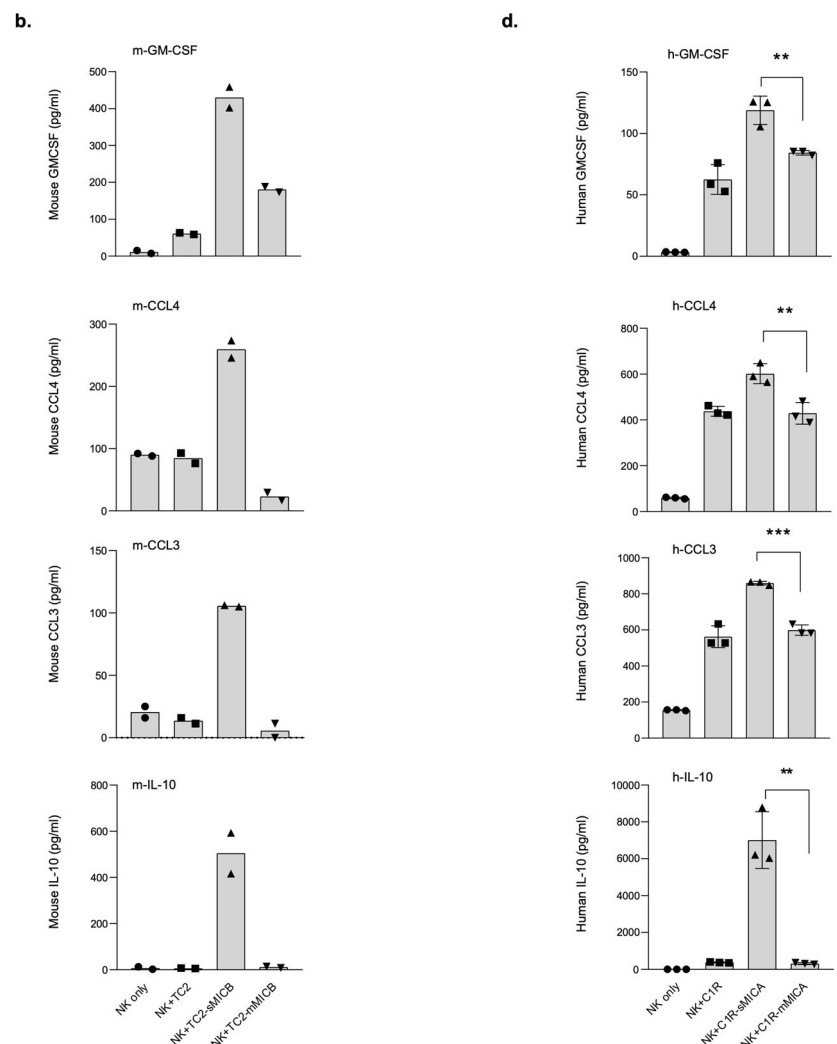

IFNγ production by both mouse and human NK cells than their counterparts cultured in presence of sMIC (Supplementary Fig. 7a–c). Different from other pro-inflammatory cytokines which are mostly regulated by activation of NF-kB, transcriptional regulation of IFNγ is directly controlled by the transcriptional factor T-bet in NK cells, similar to which in T cells[51,52], although a complex T-bet independent regulatory mechanism,

such as STAT4 and PKC-θ activation-dependent mechanisms, has been suggested[53–56]. We evaluated the expression of T-bet in human NK cells in the same co-culture experimental setting. As highlighted in Supplementary Fig. 7d, NK cells showed an increase in T-bet expression in NK cells when stimulated with mMIC, most significantly at 24 and 36 h, suggesting that the IFNγ production in NK cells, in response to mMIC stimulation is

**Fig. 3 sMIC, not mMIC, induces the enhanced expression of pro-inflammatory cytokines and chemokines. a, b** NK cells isolated from *Rag1−/−* mice and cultured in presence of IL-2 for 5 days before co-culturing with tumor cell lines TC2, TC2-sMICB, and TC2-mMICB. The supernatant was collected from the 24 h co-culture for quantitative 31-plex mouse cytokine array assay (Eve Technologies). **a** Heatmap representing significantly higher levels of pro-inflammatory cytokines and chemokines produced by mouse NK cells when stimulated with sMICB compared to stimulation with mMICB. Heatmap data are represented as a percentage change in cytokine/chemokine levels produced by NK cells stimulated with sMICB and mMICB, respectively, versus NK cells stimulated with MIC negative control TC2 cells (baseline) and normalized to 0 to 1 scale. **b** Bar graphs showing differential levels of GM-CSF, CCL4, CCL3, and IL-10 produced by mouse NK cells when stimulated with sMICB vs. mMICB. Data points in the bar graph are mean values from two independent experiments. **c, d** Primary human NK cells were co-cultured tumor cell lines C1R, C1R-sMICA, and C1R-mMICA. The supernatant was collected from the 24 h co-culture for quantitative 42-plex human cytokine array analyses (Eve Technologies). **c** Heatmap representing significantly higher levels of pro-inflammatory cytokines and chemokines produced by human NK cells stimulated with sMICA compared to stimulation with mMICA. Heatmap data are represented as percentage changes in cytokine/chemokine levels produced by NK cells stimulated with sMICA and mMICA, respectively, versus NK cells stimulated with MIC negative control C1R cells (baseline) and normalized to 0 to 1 scale. **d** Bar graphs showing significantly higher levels of cytokines and chemokines IL-10, GM-CSF, CCL4, CCL3 in the culture supernatant of human NK cells stimulated with sMICA vs mMICA. Data shown are representative of three independent experiments. **\*\*P < 0.01, \*\*\*P < 0.001** with two-tailed Student's *t* test.

regulated through T-bet. However, at 6 h, there were no significant differences in the T-bet expression across different stimulation conditions (Supplementary Fig. 7d), suggesting a complimentary T-bet independent regulatory mechanism could be employed upon initial ligand engagement. Indeed, elevated phosphorylation of PKC-θ can be seen with sMIC stimulation at 10 min as compared to mMIC stimulation as shown in Fig. 4f., suggesting a complex regulatory mechanism involving in IFNγ production upon initial NK cell activation. Together, these data support that only membrane-bound MIC can induce NK cell-protective effector function over time.

**sMIC abrogates NK cell polyfunctionality and polarizes NK cells to acquire pro-inflammatory functions**. High-strength effector polyfunctionality of CAR-T cells at the single-cell level associates with favorable clinical response[57,58]. The effector polyfunctionality refers to the ability of cells to secrete multiple (≥2) cytokines, chemokines or effector (cytotoxic) molecules that contribute towards anti-tumor responses. We thus assessed how sMIC may skew NK polyfunctionality utilizing a multi-antibody coated chip for a 32-plex single-cell cytokine response panel. Polyfunctional strength index PSI, defined as the percentage of polyfunctional cells in the sample multiplied by the intensity of the secreted cytokines, revealed that the polyfunctional NK cells secrete two categories of molecules: cytotoxic effector molecules and pro-inflammatory mediators (Fig. 6). Stimulation of ex vivo IL-2 expanded human primary NK cells with purified recombinant sMICA (rsMICA) led to a reduction in NK effector PSI, represented by a significantly decreased release of cytotoxic molecules, granzyme B and perforin (Fig. 6a). The addition of the sMIC-clearing mAb B10G5 to the culture rescued the effector PSI of NK cells. A heatmap highlights the major differences in the frequencies of an individual or combined pro-inflammatory and effector molecules secreted by NK cells when stimulated with sMIC in the presence or absence of B10G5 (Fig. 6b). Notably, polyfunctional subsets of sMIC-stimulated NK cells secreted major pro-inflammatory chemokines, MIP-1β and MIP-1α, with the minimal secretion of effector molecules. The addition of mAb B10G5 restored NK cell to the normal state of effector polyfunctionality. These data further support the finding that sMIC stimulation polarizes NK cells to a pro-inflammatory functional phenotype with a reduced cytotoxic effector function.

**Clearing sMIC effectuates adoptive NK cell therapy**. With the novel mechanistic understanding, we tested the hypothesis that antibody clearing sMIC can enhance NK cell-based therapy of MIC⁺ tumors. We adoptively transferred NK92 cells into NSG mice that were inoculated with PL12 human pancreatic cancer

cells that express MIC and shed sMIC (Supplementary Fig. 8). As shown in Fig. 7a, infusion of human NK92 cells alone had a nominal effect on tumor growth; however, infusion of NK92 cells along with the treatment of the nonblocking anti-sMIC antibody B10G5 twice-weekly resulted in significant control of tumor growth. Noteworthy, the mAb alone had no effect on tumor growth (Fig. 7a). All animals that received no treatment, NK92 cell treatment alone, or the mAb B10G5 alone reached tumor-related survival endpoint (tumor volume = 1000 mm³) by day 36 of therapy, whereas all animals that received NK92 cell therapy in combination with B10G5 mAb remained as tumor-related survivors (Fig. 7b). These data provide the proof-of-concept that clearing sMIC effectuates NK cell-based therapy for MIC⁺ tumors.

## Discussion

This study uncovers an unrecognized mechanism whereby tumor-secreted soluble factors reprograms NK cells to a pro-inflammatory dysfunctional state. This study also defined the distinctly differential functions of membrane-bound NKG2D-L mMIC and soluble NKG2D-L sMIC in editing NK cell tumor immunity, elucidated mechanisms that may underlie the controversy over the impact of NKG2D-L on NK cell tumor immunity. Mechanistically, we showed that chronic stimulation with tumor shed sMIC, activated CBM-signalosome pathway coupled with activation of PKC-θ and Fyn/ADAP in NK cells. Conversely, sustained cell membrane-bound MIC stimulation activates the PLCγ2/ Vav1/SLP-76 and ERK/JNK signaling cascades, a canonical pathway for NK cell cytolytic functions. These findings provide molecular evidence that differentiates the function of membrane MIC and soluble MIC in reprogramming NK cells. These molecular findings are consistent with a *de novo* NK cell transcriptional activity and enriched transcripts of pro-inflammatory genes upon sMIC stimulation and in vivo single-cell RNAseq to associate tumor progression with high and low content of sMIC. Further, in vivo NK cell-based therapeutic study provided the proof-of-principle that targeting sMIC can effectuate NK cell-based cancer immunotherapy. Together, we uncovered a previously unrecognized immunosuppressive mechanism conferred by sMIC and an undescribed aspect of NK cell functional plasticity in the tumor microenvironment as depicted in Fig. 8. This study paved the way for targeting sMIC to enhance NK cell-based cancer immunotherapy.

MIC family polypeptides are the most characterized human NKG2D-L in relevance to cancer and other pathogenic conditions[59]. While expression of MIC on tumor cells represents an innate mechanism to trigger active anti-tumor responses, controversial clinical conclusions relating to this pathway have been drawn based on limited mechanistic understandings[14,15,60].

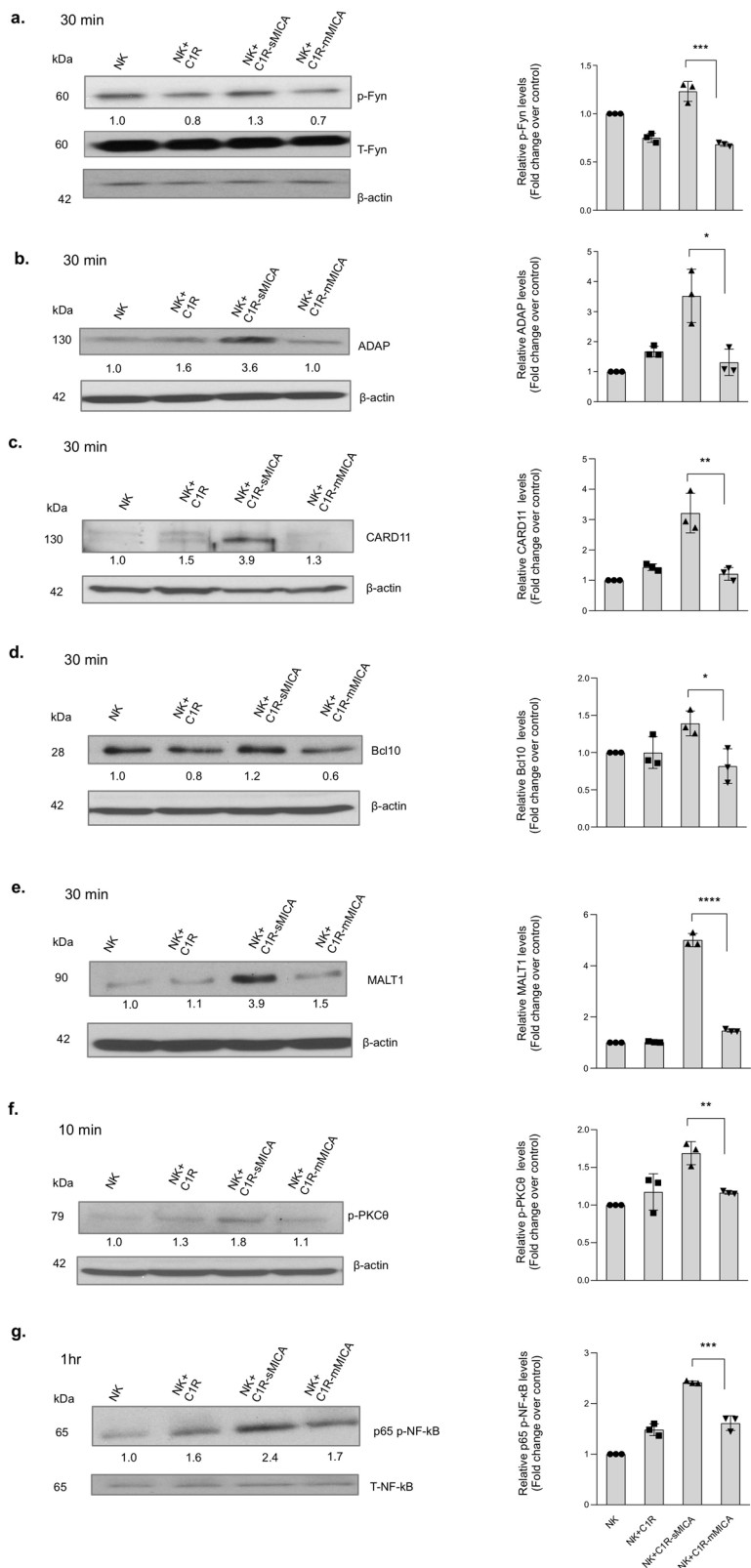

One possibility for the controversy to arise could be the inability to distinguish the effects of sMIC versus cell membrane MIC in these studies. We have now demonstrated that the soluble form of NKG2D-L sMIC reprograms NK cells to a pro-inflammatory and pro-tumorigenic functional status via activating the CBM-signalosome pathways to secrete pro-inflammatory cytokines, such as CCL3 and CCL4. These mediators are known to promote a pro-tumorigenic inflammatory tumor microenvironment through recruiting inflammatory myeloid cells, and macrophages[61–63]. Indeed, it was shown that sMIC[hi] tumors had a significantly higher accumulation of myeloid suppressor cells[64]. It is conceivable that, with the amplified inflammatory tumor microenvironment and concurrent immune-suppressive effects induced by sMIC, tumors could override the protective immunity

**Fig. 4 sMIC preferentially activates pro-inflammatory signaling pathways in NK cells through the activation of PKC-θ and ADAP and the downstream CBM-signalosome pathway.** Primary human NK cells were co-cultured with tumor cell lines C1R, C1R-sMICA, and C1R-mMICA that were respectively adherent to poly-L-lysine-coated plates for indicated timepoints. At indicated stimulation timepoints, NK cells were collected, pelleted, washed, lysed in RIPA buffer containing protease inhibitor cocktail and phosphatase inhibitor, and analyzed with immunoblotting. **a** Fyn phosphorylated at Tyr530 (p-Fyn); **b``** adaptor signaling molecule ADAP; **c–e** components of CBM signalosome CARD11, Bcl10, MALT1; **f** protein kinase C-theta (PKC-θ) phosphorylated at Thr538 (p-PKC-θ); **g** NF-κB p65 phosphorylated at (Ser536). Individual phospho-proteins, their respective total proteins, and β-actin were presented from the same blots. The blots were first probed for phospho-proteins, followed by stripping and re-probing for respective total proteins and subsequently for β-actin. The numbers represent quantified values after normalization (and represented as relative to NK only condition). The blots (left panel) shown are representative of three independent experiments. The bar graphs (right panel) show the quantitative data of relative protein levels and are represented as the fold change over NK cell controls. All phosphorylated proteins were normalized to total proteins and adaptor proteins were normalized to β-actin loading control. Quantification was performed using Image J software. *$P < 0.05$, **$P < 0.01$, ***$P < 0.001$, and ****$P < 0.0001$ (Student's $t$ test; two-tailed).

provided by residual tumor cell surface MIC. Noteworthy, tumor-shed or recombinant sMIC contains the ectodomain (α1α2α3) of full-length MIC and that NKG2D recognizes the same ectodomain of α1 and α2 of sMIC and membrane-bound MIC[65,66]. It is intriguing how the same recognition of sMIC and mMIC by NKG2D could lead to a partitioned NK cell-intrinsic signaling and thus polarized functional reprogramming. One conceivable explanation for the outcome could be that a potential sustained and high-magnitude stimulation by sMIC without immune synapse formation results in deregulation of NKG2D signaling cascade[49]. How the intricate biophysical interactions lead to distinct biochemical and functional outcomes warrants further investigations.

NK cells are critical components in controlling tumor growth and metastasis[67]. NK cell-based cancer therapies are currently at the forefront of development. However, to date only limited efficacy has been achieved[1,28]. It was reported that tumor-infiltrating NK cells often acquire altered phenotype with diminished cytotoxic function in response to the immune-suppressive tumor microenvironment[68–70]. Given that MIC is prevalently expressed by almost all solid tumors[9,10], our study suggests that tumor-shed sMIC could propagate the immune-suppressive tumor microenvironment by skewing NK cell function towards pro-tumorigenic chemokine secretion and subsequent amplifying the immune-suppressive milieu. A critical question is whether clearance of sMIC can reverse the malfunction of NK cells. Indeed, our current data demonstrate that antibody clearance of sMIC restores NK cell cytotoxicity against target cells expressing mMIC. Our current finding is consistent with our in vivo findings that therapy with the sMIC-clearing antibody heightens tumor NK cell cytotoxic function and reprograms tumor microenvironment[30]. RNAseq data from our current study further confirmed that an antibody clearance of sMIC reprograms dysfunctional NK cells to the cytotoxic effector state. This concept was further proven with the current adoptive NK cell therapeutic studies. Therefore, it is suggestive that sMIC, either intratumorally or in circulating, should be evaluated when considering NK cell-based immunotherapy of cancer and that antibody clearance of sMIC holds promise for enabling its therapeutic effect.

In summary, our current study uncovered a previously unde-scribed immune-modulatory effect of human soluble NKG2D-L sMIC in reprogramming NK cell to a dysfunctional pro-inflammatory phenotype. In a broad aspect of immune regulation beyond tumor immunity, our findings will help to understand the potential pathological role of NK cells in exacerbating inflammatory disorders, for instance of rheumatoid arthritis where sMIC is abundantly present in the synovial fluid of patients[71–73]. Funda-mentally, we have delineated a previously undescribed mechanism to askew NK cell function plasticity in the tumor microenvironment and molecular pathways whereby soluble NKG2D-L sMIC polarizes NK cells to the pro-inflammatory function. Our study ascertained

the potential of targeting sMIC to improve NK cell-based cancer immunotherapy.

## Methods

**Mice and the anti-sMIC/MIC monoclonal antibody B10G5.** All the mice used in the study were housed and bred under specific pathogen-free (SPF) conditions in the animal facility at Northwestern University in accordance with Institutional Animal Care and Use Committee (IACUC) approval. Both male and female Rag1$^{-/-}$ mice were used for the isolation of splenic NK cells. The generation and characterization of bi-transgenic TRAMP/MICB mice have been previously described[6]. Generation and characterization of the anti-sMIC/MIC monoclonal antibody B10G5 have been previously described[6,30].

**Primary NK cell isolation.** Primary mouse NK cells were isolated from the spleens of Rag1$^{-/-}$ mice of 8–12 weeks old. Briefly, after the lysis of red blood cells (RBC) using ammonium-chloride-potassium (ACK) lysis buffer, single-cell suspension of splenocytes was seeded for 2 h at 37 °C to remove adherent cell populations. Non-adherent NK cells were collected and cultured in presence of 1000 IU/ml of recombinant human IL-2 (TECIN$^{TM}$ Teceleukin, Bulk Ro 23-6019, National Cancer Institute) and 50 μM beta-mercaptoethanol for 5 days. The purity of NK cells was confirmed to be >98%.

For primary human NK cell preparation, frozen peripheral blood mononuclear cells (PBMCs) (Stem cell technology, 70025.3) from healthy donors were used as the source of NK cells. Frozen vials of PBMCs were thawed and rested overnight in RPMI 1640 media supplemented with 10% FBS, 1% penicillin–streptomycin (Gibco, 15070063), and 100 IU/ml of recombinant human IL-2. After overnight resting, NK cells were enriched and expanded from the PBMCs using CellXVivo$^{TM}$ Human Cell Expansion Kit (R&D systems, CDK015) according to the manufacturer's instructions. NK cells with the purity of >95% CD3$^{-}$CD56$^{+}$ were used for the experiments.

**sMIC and mMIC-expressing tumor cell lines.** Mouse prostate tumor cell line TRAMP-C2 (TC2) and human B-cell lymphoblast tumor cell line C1R were, respectively, transduced to express recombinant soluble (sMICB) and sMICA or the shedding-resistant form of mMICB and mMICA (previously referred to as MICB.A2 and MICA.A2) using retroviral transduction system, as described previously[32,33]. All tumor cell lines were cultured in RPMI 1640 media supplemented with 10% FBS and 1% penicillin–streptomycin.

**Cell co-culture and cytokine array assay.** Primary mouse NK cells were co-cultured with mouse prostate tumor TC2 cell lines expressing soluble and membrane-restricted MIC ligands, TC2-sMICB and TC2-mMICB, respectively, at the ratio as indicated. Similarly, primary human NK cells were co-cultured with human B-cell lymphoblast cell line C1R expressing soluble and membrane-restricted MIC ligands, C1R-sMICA and C1R-mMICA, respectively, at the indicated ratio. After 24 h co-culture, the supernatant was collected for the multiplex cytokine array assay (Eve Technologies, Calgary, AB, Canada).

**Western blotting.** Primary human NK cells (4–5 million cells for each experimental condition) were co-cultured with C1R, C1R-sMICA, and C1R-MICA.A2 cells at 1:1 ratio for indicated timepoints. Six-well plates (Corning, 3450) were used for the experiments. Wells were coated with poly-L-lysine (Sigma, P4707) for 5 min, excess solution was removed, rinsed thoroughly with sterile tissue culture grade water. After drying wells for 2–4 h, tumor cells were seeded in the wells and allowed to adhere for 3–4 h before the addition of NK cell suspension. At indicated timepoints of stimulation, NK cells present in the culture suspension were collected from the wells and separated from the adherent tumor cells. The collected NK cells were then pelleted down and washed twice with ice-cold PBS. For plate-bound sMIC conditions, recombinant sMICA (Sino Biologicals, HPLC-12302-H08H) was diluted in sterile PBS (1 μg/ml) and coated on the wells of a sterile six-well plate and stored at 4 °C overnight. Prior to seeding of the NK cells, the wells were rinsed

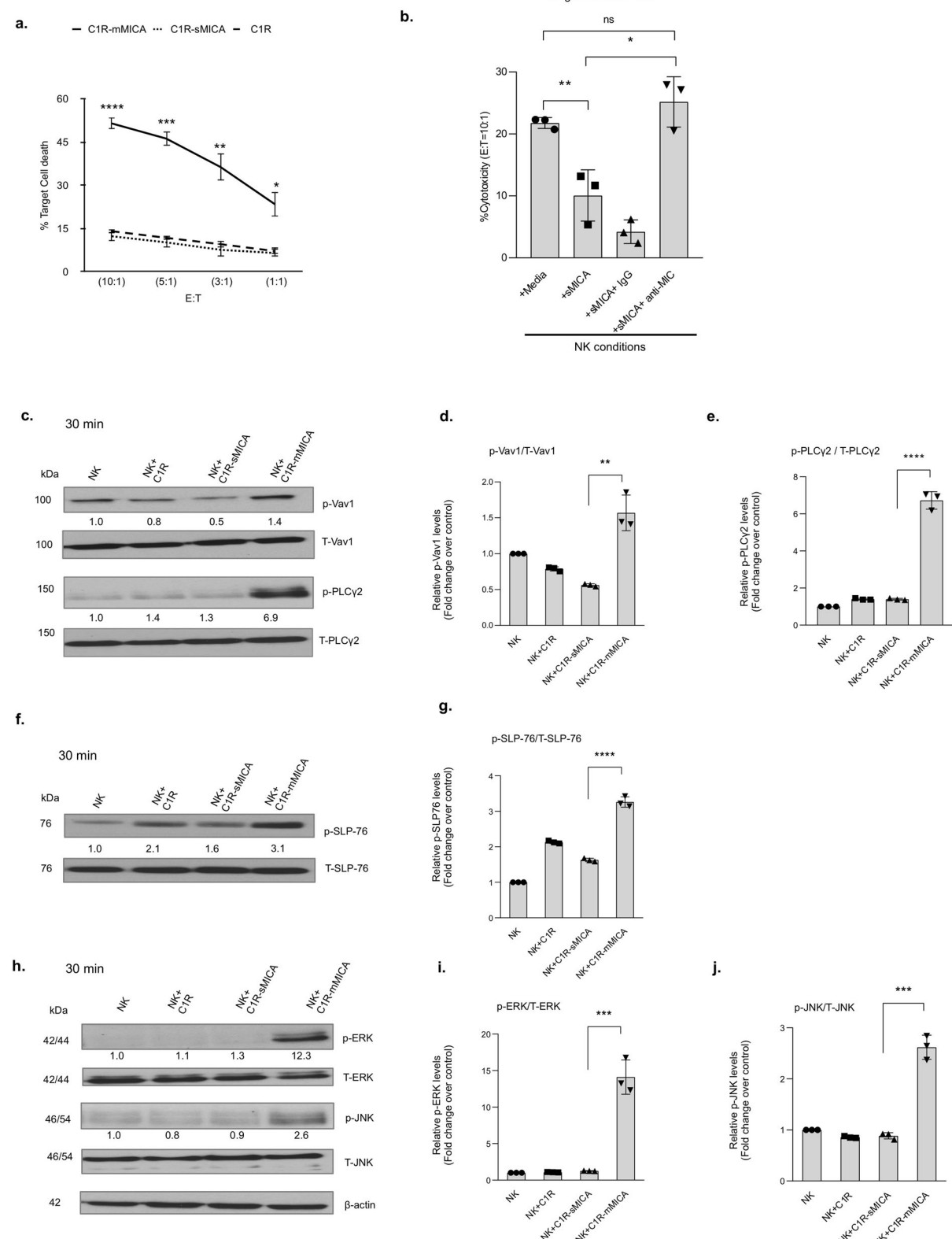

twice with sterile PBS. Cell lysates were prepared using RIPA buffer supplemented with protease inhibitor cocktail (Cell Signaling Technology, 5871) and phosphatase inhibitor 1 mM sodium orthovanadate. Total protein quantification of cell lysates was performed using BCA protein assay kit (Pierce, 23225). In total, 30 μg of protein for each sample was used to load SDS-PAGE, followed by transferring to PVDF membranes. The membrane was blocked for 1 h using 5% nonfat milk at room temperature and then incubated with respective primary antibodies prepared in 5% BSA in 1× TBST buffer at 4 °C overnight. The membranes were washed three times with TBST (1X TBS + 0.1% Tween-20) and then incubated with secondary antibody anti-rabbit/mouse IgG, horseradish peroxidase (HRP)-linked (Cell Signaling, 7076) prepared in 5% nonfat milk in 1× TBST buffer for 1 h at room temperature. After washing with TBST, proteins were detected using enhanced chemiluminescence western blot substrate (Pierce, 32209). The following primary antibodies were purchased from Cell signaling technology (CST): phospho-PKC$_\theta$ (Thr538) (CST, 9377), PKC$_\theta$ (CST, 13643), MALT1 (CST, 2494), Bcl10 (CST, 4237), CARD11 (CST, 4435), Fyn (CST, 4023), phospho-NF-κB p65 (Ser536)

**Fig. 5 Membrane-bound, but not soluble MIC, preferentially enhances NK cell cytotoxicity and associated signaling pathways. a** Cytotoxicity of human NKL cell line cultured with labeled target cells C1R, C1R-sMICA, C1R-mMICA at different E/T ratios for 4 h, as assessed by flow cytometry analyses of 7-ADD for tumor cell death. **b** Cytotoxicity of primary human NK cells against C1R-mMICA with stimulation conditions of sMICA (100 ng/ml) alone or in presence of B10G5 (5 μg/ml) assessed by lactate dehydrogenase (LDH) release. Immunoblot analysis to evaluate the activation of (**c–e**) Vav1 phosphorylated at Tyr 160 (p-Vav1) and PLCγ2 phosphorylated at Tyr 1217 (p- PLCγ2); **f**, **g** SLP-76 phosphorylated at Tyr 128 (p-SLP-76); and **h–j** ERK phosphorylated at Tyr 202/Tyr204 (p-ERK) and JNK phosphorylated at Thr183/Tyr185 (p-JNK) in human NK cells at the indicated condition and timepoints. Phosphorylated proteins were normalized to respective total proteins. Individual phospho-proteins, their respective total proteins, and β-actin were presented from the same blots. The blots were first probed for phospho-proteins, followed by stripping and re-probing for respective total proteins and subsequently for β-actin. The numbers represent quantified values after normalization and are represented as relative to NK only control conditions. The blots shown are representative of three independent experiments. The bar graphs show the quantitative data of relative protein levels and are represented as fold change over NK cell controls. All phosphorylated proteins were normalized to total proteins. Quantification was performed using Image J software. *P < 0.05, **P < 0.01, ***P < 0.001, and ****P < 0.0001 (Student's t test; two-tailed).

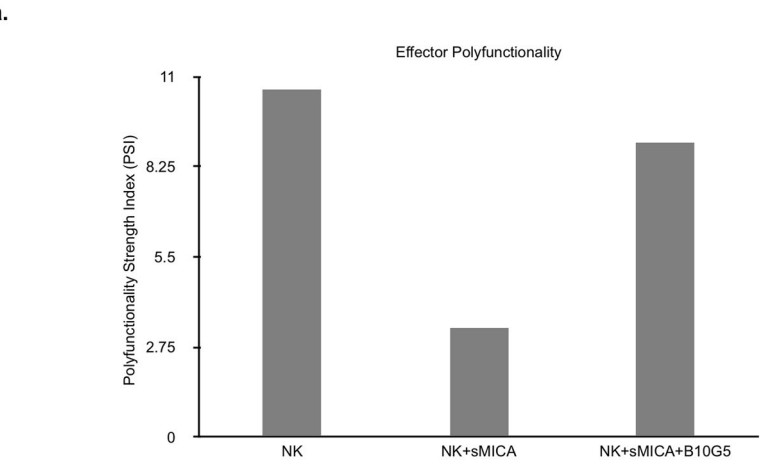

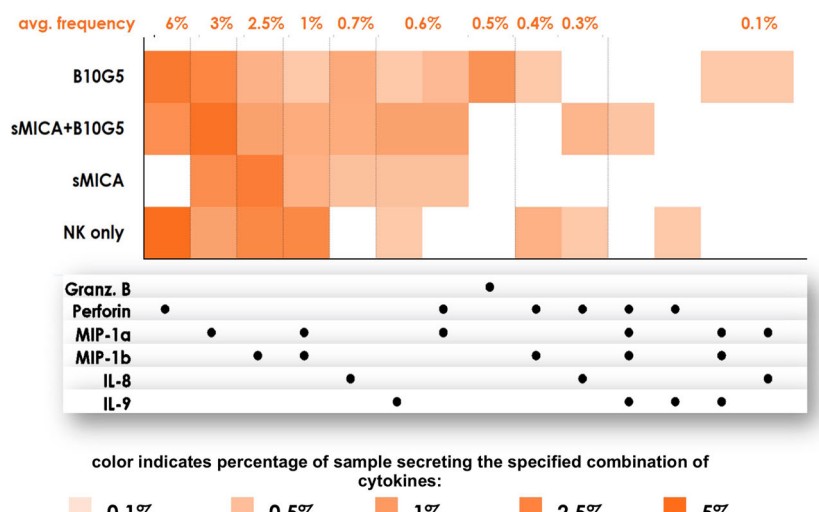

**Fig. 6 sMIC abrogates the effector polyfunctionality of NK cells and polarizes NK cells to possess a pro-inflammatory phenotype.** Primary human NK cells were cultured with recombinant soluble 100 ng/ml of sMICA for 24 h in the absence and presence of the sMIC-clearing mAb B10G5. In total, 30 μL of cell suspension was loaded onto an IsoCode chip with a 32-plex antibody array for the measurement of secreted proteins. Polyfunctional Strength Index (PSI) was calculated as the percentage of polyfunctional cells in the sample, multiplied by the intensities of the secreted cytokines. **a** Bar graph representing the effector polyfunctionality strength index (PSI) of NK cells. **b** Single-cell functional heatmap illustrating differences in the single-cell secretion frequencies of various individual cytokines or combinations of cytokines. Each sample or condition is a row of the heatmap, while each column corresponds to the secretion of a specific cytokine or a combination of cytokines. The dot(s) in each column indicate which cytokine(s) are represented by that column. Secretion frequency is denoted by the shade of orange color; the darker the orange color, the more frequently the corresponding cytokine(s) were secreted by a specific sample or condition, as a percentage of profiled single cells.

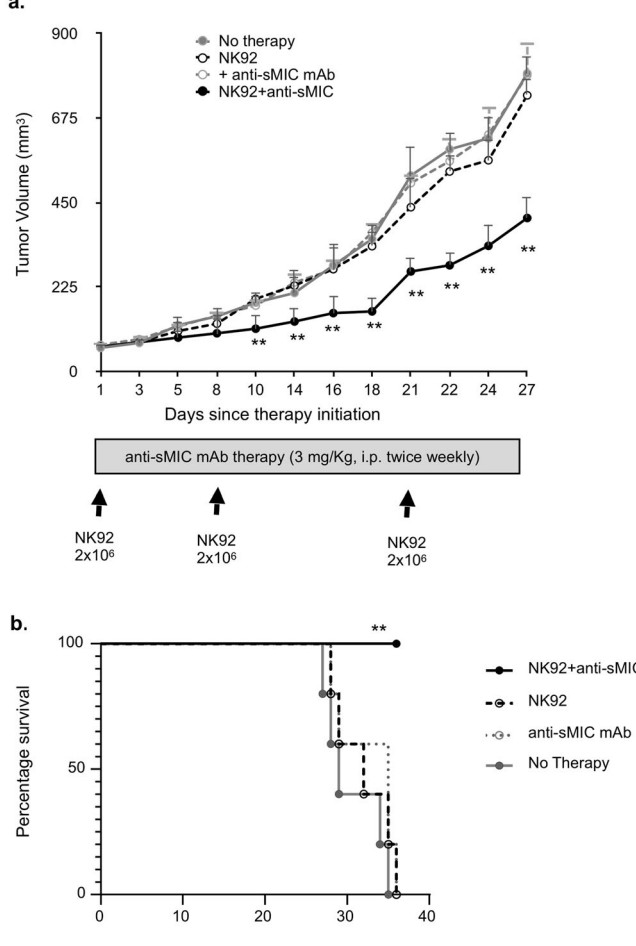

**Fig. 7 Co-administration of sMIC-clearing nonblocking antibody effectuate NK92 cells controlling the growth of MIC+ tumors in vivo.**
PL12 pancreatic cancer cells were injected into NSG mice ($1 \times 10^6$ cells/mouse). When tumors reached 50–100 mm³ in size, mice were randomized into four therapy groups ($n = 6$/group) as indicated. Tumor burden of 1000 mm³ was defined as tumor-related survival endpoint. The study was ended when all animals in the control group, NK92 only group, and anti-sMIC treatment groups had tumor burden of 1000 mm³. **a** Average tumor growth curve in each treatment group. Note the growth curve only plotted within the period when no animals reached the tumor-related survival endpoint. **b** Kaplan–Meier survival curve. **$P = 0.0007$ compared to NK92 treatment group with Mantel–Cox test.

(CST, 3033), NFκB p65 (CST, 4764), phospho-ERK(1/2) (Thr202/Tyr204) (CST, 9101), phospho-JNK (Thr183/Tyr185) (CST, 9251), JNK (CST, 9252), phospho-PLCγ2 (Tyr1217) (CST, 3871), PLCγ2 (CST, 3872), Vav1 (CST, 2502), SLP-76 (CST, 70896), β-actin (CST, 3700), DAG lipase β (CST, 12574), phospho-IP3 (CST, 8548), phospho-Lck (CST, 2751), Lck (CST, 2752), GAPDH (CST, 2118). Anti-ADAP antibody was obtained from Millipore Sigma (07-546). Phospho-Vav1 (Tyr160) was from Invitrogen (44-482), phospho-SLP-76 (Tyr128) was from BD Biosciences (558367) and ERK(1/2) antibody was obtained from Santa Cruz biotechnology (sc-514302). Anti-GFP antibody was from Abcam (ab13970).

**Cytotoxicity assay**. For flow cytometry-based cytotoxicity assay, target cells were first labeled with a cell trace far-red dye (Invitrogen, C34572) for 30 min and seeded in a 96-well round-bottom plate (10,000 cells/well). Effector NKL cells were added at indicated ratios. After brief centrifugation, cells were co-incubated for 4 h. followed by being washed twice with cold PBS and stained with 7-AAD dye (BD biosciences, 559925) for 15 min for flow cytometry analyses. Percentage of cell death was calculated by [(7-AAD⁺ Target cells)/(total target cells)] × 100. For lactate dehydrogenase (LDH) cytotoxicity assay, experiments were conducted according to the manufacturer's protocol (Thermofisher Scientific, 88953). Briefly, target cells were seeded in a 96-well plate U-bottom plate with triplicates in each

experimental condition. For data in Fig. 5b, 100 ng/ml of recombinant sMICA (Sino biologicals, 12302-H08H) or sMICA (100 ng/ml) in the presence of the anti-sMIC antibody B10G5 (5 µg/ml) or control IgG antibody was used. Effector primary human NK cells were added to the plate at indicated E/T ratio and co-incubated at 37 °C for 4 h. Conditioned medium was collected and transferred to a new 96-well flat-bottom plate and incubated with the reaction mixture for 30 min at room temperature. The reaction was terminated by adding a stop solution. The absorbance was measured at 490 and 680 nm on a spectrophotometer (Spectramax, Molecular Devices). The cytotoxicity of NK cells was calculated as: % cytotoxicity = [(experimental value − effector cells spontaneous control − target cells spontaneous control)/(target cell maximum control − target cells spontaneous control)] × 100.

**Real-time quantitative PCR (qRT-PCR)**. RNA was isolated using RNeasy Mini kit (Qiagen, 74106) and then reverse transcribed into cDNA using iScript cDNA synthesis kit (Bio-rad, 1708891) following the manufacturer's instructions. mRNA expression was examined using SsoAdvanced SYBR green dye (Bio-rad 1725274) based PCR amplification and Bio-Rad CFX96 real-time detection system. All primer sequences are listed in Supplementary Table S2.

**RNA-sequencing sample preparation and analysis**. Mouse NK cells were isolated from Rag1⁻/⁻ mice, cultured in presence of IL-2 for 5 days (as described above) with a purity of more than 98% NK1.1⁺ cells, and stimulated with recombinant soluble MICB (Sino biologicals, 10759-H08H) in the absence or presence of B10G5 for 18 h. Cells were collected. RNA was extracted using RNeasy Mini kit (Qiagen, 74106). Mouse NK cell RNA-seq libraries were prepared at genomics core, UCLA using KAPA stranded RNA-seq kit. Sequencing was performed on Illumina Hiseq3000 for a single read 50 bp run. The quality of DNA reads, in fastq format, was evaluated using FastQC (https://www.bioinformatics. babraham.ac.uk/projects/fastqc/). Adapters were trimmed, and reads of poor quality or aligning to rRNA sequences were filtered. The cleaned reads were aligned to the *Mus musculus* genome (mm10) using STAR program. Normalization and differential expression were determined using DESeq2. The cutoff for determining significantly differentially expressed genes was a fold change cutoff of 2 and FDR-adjusted $P$ value <0.05.

Human PBMC enriched NK cells were stimulated with recombinant sMICA (Sino biologicals, 10759-H08H and R&D systems, 1300-MA-050) in the absence or presence of the anti-MIC mAb B10G5 for 18 h. RNA was extracted using RNeasy Mini kit. Human NK cell RNA-seq libraries were prepared at Novogene Inc. using NEB-Next Ultra II RNA library prep kit (New England Biolabs Inc., E7775). The sequencing was performed on Hiseq3000 for a single end 50 bp run. After quality control with FastQC, trimming of adaptors and filtering of reads aligning to rRNA, the raw reads were aligned to the human genome HG19 using the STAR aligner version 2.7.3a. Read counts for each gene were computed with the gene annotations from UCSC using HTSeq. Differential expression of mRNA level between different groups was analyzed using DESeq2 version 1.28.0. The cutoff for determining significantly differentially expressed genes was the fold change of 2 with FDR-adjusted $P$ value <0.05.

**Single-cell multiplex cytokine profiling of human NK cells**. Frozen primary human NK cells (enriched from PBMCs) were recovered in complete RPMI media with IL-2 (10 ng/ml, Biolegend) at a density of $1–5 \times 10^6$ cells/ml at 37 °C, 5% $CO_2$ overnight. Viable NK cells were isolated using Ficoll centrifugation. Cells were cultured with sMICA 100 ng/ml (R&D systems, 1300-MA-050) in the presence and absence of B10G5 (100 ng/ml) for 24 h. In all assays, 30 µL of cell suspension was loaded at a density of $1 \times 10^6$ cells/ml into IsoCode chip (IsoPlexis, Branford, CT) with a 32-plex antibody array for measurement of secreted proteins. The single-cell polyfunctional profiles of NK cells were evaluated with a focus on pro-inflammatory and cytotoxic functional groups. Polyfunctional Strength Index (PSI) was calculated as the percentage of polyfunctional cells in the sample, multiplied by the intensities of the secreted cytokines. Data analysis and heatmap generation were conducted using Isoplexis' software.

**Single-cell RNA-sequencing sample preparation and data analysis**. After aseptic harvest, tumors were finely minced using scalpel blade. The tumor slurry was suspended in cold HBSS (Thermofisher, 14170120) and centrifuged at $300 \times g$ for 5 min. RBCs were lysed using ACK lysis buffer followed by washing with PBS. To obtain a single-cell suspension, an enzymatic digestion procedure was carried out in which the tumor slurry was suspended in 5 ml of a 1:10 dilution of collagenase (Stem Cell Technology, 07912) in DMEM/F12 media plus 10% FBS at 37 °C for 30 min with occasional mixing to help the digestion process. After centrifugation at $350 \times g$ for 5 min, the cell pellet was resuspended in 3 ml pre-warmed dispase (5 U/ml) plus 300 µl of DNase I (1 mg/ml) and triturated vigorously for 1 min. The suspension was passed through 100-µm cell strainer while mashing simultaneously with the back of the syringe plunger followed by centrifugation at $300 \times g$ for 5 min. After washing, the cell suspension was stained with fixable viability dye eFluor 450, followed by surface staining for anti-mouse CD45 (Biolegend, 103106) and sorting for CD45⁺ cells on BD FACS Aria III instrument. After sorting, cells were counted using a hemocytometer confirmed that viability

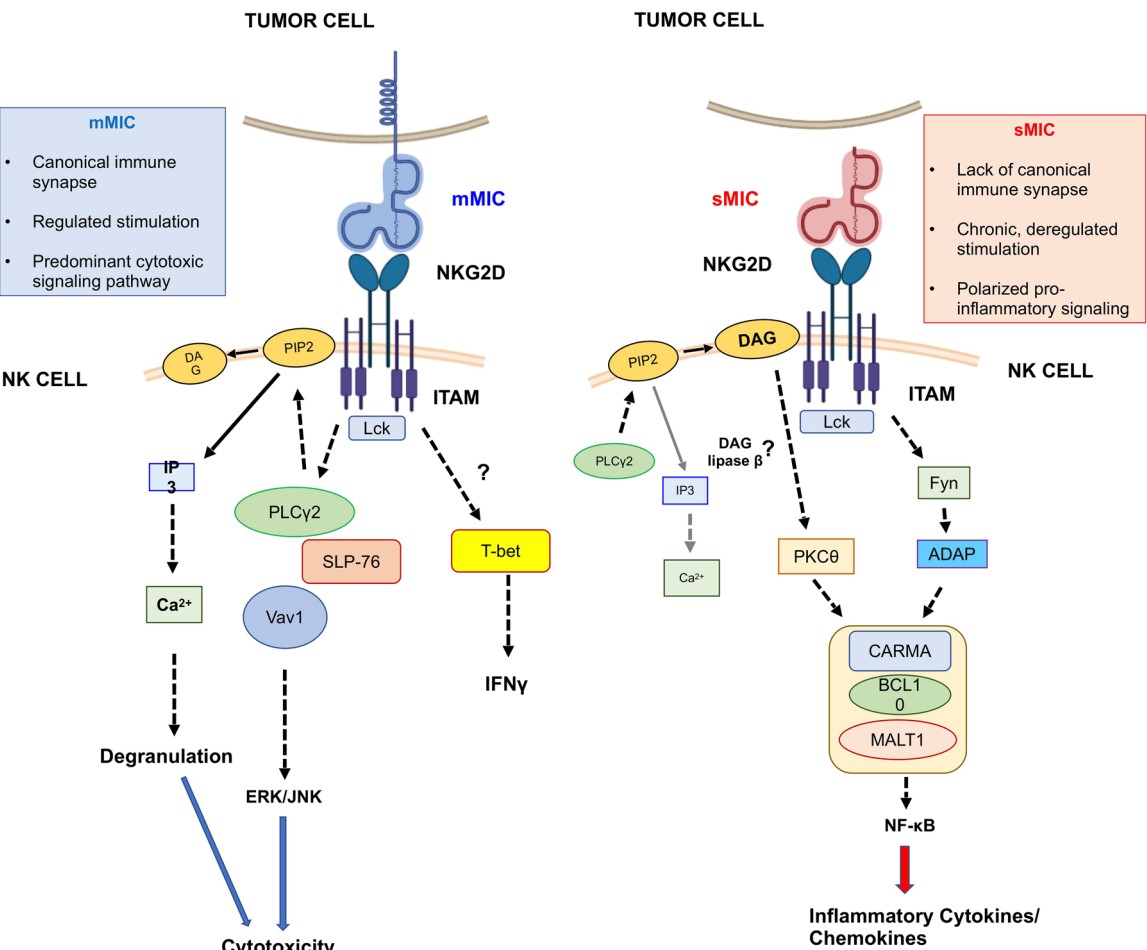

**Fig. 8 Proposed model of the molecular pathways whereby mMIC and sMIC differentially regulate NK cell function.** Engagement of the receptor NKG2D to the two forms of MIC ligand, mMIC, and sMIC leads to distinct signaling cascades resulting in activation of NK cell cytotoxicity and inflammatory cytokine/chemokine production, respectively. mMIC-NKG2D interaction delivers a regulated stimulation through canonical immune synapse formation and selectively activates PLCγ2/SLP-76/Vav-1 and ERK/JNK signaling axis for NK cell cytotoxicity and upregulation of T-bet for IFNγ secretion. Conversely, the lack of canonical immune synapse formation upon sMIC-NKG2D interaction delivers a chronic and deregulated stimulation that results in activation of CBM-signalosome-mediated pro-inflammatory pathways. The activation of CBM-signalosome pathways is associated with persistent activation of Fyn/ADAP and PKC-θ pathways. The ITAM immediate downstream molecule Lck is required for both mMIC- and sMIC-stimulated events.

was >85%. Cells were loaded to 10× chromium machine (10× genomics), in which the library preparation procedure was conducted according to the standard protocol of the Chromium single-cell 3' kit, V3 chemistry. Sequencing was performed on Hi-Seq 4000 system (Illumina, 100-bp paired-end protocol).

The kallisto | bustools (kallisto v0.46.0 and bustools v0.39.1) workflow was used to pre-process the single-cell RNA-seq unfiltered feature-barcode matrix generated by CellRanger (version 3.1.0; 10× Genomics)[74]. Kallisto index was built with reference transcriptome GRCm38 version 97. For each sample, the emptyDrops method implemented in DropletUtils R package was applied to detect and remove empty droplets[75]. The data analysis was performed using the Seurat R package on the UMI count gene-by-cell matrix[76]. Low-quality cells were filtered out if meeting one of the following criteria: (1) with genes detected in less than three cells; (2) with less than 200 genes expressed; (3) with >3 standard deviations above the mean number of genes detected; (4) with >5% of their UMI counts mapping to MT genes. The UMI counts were normalized using sctransform method[77]. Potential doublets were detected and removed with DoubletFinder R package[78]. Data integration using Harmony was performed to minimize the batch effects caused by the unwanted technical variations[79]. The highly variable genes were subsequently selected for principal component analysis (PCA). The first 30 principal components of the z-normalized data were used for clustering cells using Seurat k-nearest neighbors clustering with resolution 1.6. Cell populations were identified for each cluster by examining the expression of marker genes and the top differentially expressed genes in each cluster. Cells were visualized by UMAP on the same 30 principal components. Cells from NK clusters were extracted and applied to the HIPPO method to identify NK cell subtypes[31]. MAST algorithm implemented in FindMarker function from Seurat package as applied to identify differentially expressed genes in NK cell subtypes between PD and WD tumors. Gene set enrichment analysis was performed using the functions implemented in clusterProfiler R package[80]. The threshold for a significantly enriched gene set is FDR controlled P value of 0.25.

**Tumor processing and ex vivo stimulation assay.** Tumors were harvested from TRAMP/MICB mice after euthanization. One gram of tumor from all the mice was weighed and used for further processing. Tumors were finely minced using scalpel blade and then directly mashed with the back of syringe plunger using a 100-μm strainer. The tumor single-cell suspension was used for functional analysis of TILs. Cells were stimulated with 50 ng/ml phorbol myristate acetate (PMA) and 500 ng/ml ionomycin at 37 °C for 3 h. Golgi plug (1:1000) and golgi stop (1:1500) (BD Biosciences, 555029, 554724) were added to the cells in the last 2 h of stimulation. IFNγ, granzyme B, and TNFα production were analyzed by intracellular staining with BD cytofix/cytoperm kit (BD Biosciences, 554714) following the manufacturer's instructions.

**Flow cytometry.** For co-culture experiments, primary mouse and human NK cells were co-cultured with TC2 and C1R cell lines expressing soluble and membrane-bound MIC ligands respectively for indicated timepoints at a ratio of 1:1 (100,000 cells each). GolgiPlug™ (1:1000, BD Biosciences) and GolgiStop™ (1:1500, BD Biosciences) were added in the last 2 h of co-culture. For the functional analysis of tumor-infiltrating NK cells from TRAMP/MICB mice, after processing the tumors as described in the section above, tumor single-cell suspensions were seeded and activated as described above. Cells were washed twice with PBS, stained with fixable viability dye eFluor 450 (ebioscience, 65-0863-14), followed by staining for cell surface markers. For intracellular staining, after staining for cell surface markers, cells were fixed and permeabilized using BD cytofix/cytoperm kit (BD Biosciences, 554714). Staining for T-bet was performed using True Nuclear transcription factor

buffer set (Biolegend, 424401) following the manufacturer's instructions. The following fluorochrome-conjugated antibodies were obtained from Biolegend: anti-human CD56 (362508), anti-human CD3 (300318), anti-human IFNγ (502528), anti-human T-bet (644810), anti-mouse CD3 (100222), anti-mouse CD45 (103128), anti-mouse IFNγ (505826), anti-mouse NK1.1(108710), anti-mouse TNFα (506322), anti-mouse Granzyme B (515403). BD Fortessa cytometer was used for flow cytometry data collection. FlowJo software (Tree Star) was used for the data analysis.

**in vivo studies**. All described studies were approved by the Institutional IACUC review committee. Cohorts of 8–10 weeks old NSG mice (male and female) were subcutaneously inoculated with $1 \times 10^6$ cell/mouse human pancreatic cancer cell PL12 (provided by Dr. Nicholas Cacalano, UCLA Jonsson Comprehensive Cancer Center[81]). When tumors reached 50–75 mm$^3$ in size, animals were randomized into four therapeutic groups with equal numbers of male ($n = 3$) and female ($n = 3$) in each group. The four therapeutic groups are: (1) no therapy; (2) receiving $i.v.$ injection of NK92 cell line at $2 \times 10^6$ cells/dose as indicated in Fig. 7a for dosing schedule; (3) receiving $i.p.$ injection of the anti-sMIC mAb B10G5 twice weekly at the dose of 3.0 mg/Kg; (4) a combinational therapeutic schema of group (2) and group (3). Tumor growth and body weight were monitored twice weekly. Tumor volume of 1000 mm$^3$ was defined as tumor-related survival endpoint for Kaplan–Meier survival analyses. The study was ended when all animals in groups 1–3 reached defined tumor-related survival endpoint.

**Statistical analysis and reproducibility**. All the statistical data were indicated as mean ± SEM. All in vitro co-culture and functional data are from at least three independent experiments unless stated otherwise in the figure legends. A standard student's $t$ test was used to analyze the significance of differences observed between groups. Mantel–Cox test was used for comparison of survival in Fig. 7. $P$ values of <0.05 were considered to be statistically significant. GraphPad Prism software was used to conduct statistical analyses.

## Ethics approval and consent to participate
All animal studies were approved by the Institutional Animal Care and Use Committee (IACUC) committee of Northwestern University.

**Reporting summary**. Further information on research design is available in the Nature Research Reporting Summary linked to this article.

## Data availability
All source data underlying the graphs and charts presented in the main figures are provided in the excel format as Supplementary Data 1. Full-sized uncropped raw images of western blots are summarized in Supplementary Figs. 9 and Fig. 10. Any remaining information related to the data generated or analyzed in this study is available from the corresponding author upon reasonable request.

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

## Acknowledgements

This work was supported by NIH/NCI grants 1R01CA208246, 1R01CA204021, and 1R01CA212409 (to J.D.W.). We sincerely thank Dr. Ju Wu for his assistance in organ collection for the single-cell RNA-sequencing experiment. We sincerely thank Mr. Nitin Kak for his technical assistance in generating part of the final graphs of the single-cell RNA-sequencing data.

## Author contributions

J.D.W. and P.D. conceived and designed the research study. P.D., F.B., and J.Z. performed the experiments and acquired the data. P.D., F.B., Z.J., L.H., S.Q., J.Z., J.R., S.H., S. M., and J.D.W. analyzed and interpreted the data. P.D., J.D.W., and D.A.W. wrote and edited the manuscript. P.D., J.D.W., D.A.W., F.B., Z.J., L.H., S.Q., J.Z., J.R., S.H., and S.M. reviewed the manuscript. J.D.W. supervised the study.

## Competing interests

J.D.W. is the inventor of the B10G5 antibody with international patents. J.Z. is employed by and has equity ownership in IsoPlexis; S.M. is cofounder of, has equity ownership in, and holds patents with IsoPlexis. The remaining authors declare no competing interests.
