## [Peer Review File · Communications Biology]

Reviewers' Comments:

Reviewer #1:

Remarks to the Author:

In this manuscript Dhar et al presented evidence for the role of sMIC in inactivation of NK cell function. They presented evidence that sMIC reprogrammed NK cell to secrete pro-tumorigenic cytokines with diminished cytotoxicity and decreased polyfunctionality. They further demonstrated that antibody clearing sMIC restored the NK cell function. sMIC was also found to selectively activate the CBM-signalosome inflammatory pathways in NK cells. In addition, tumor cell membrane-bound MIC (mMIC) stimulated NK cell function through PLC2γ2/SLP-76/Vav1 pathway. In vivo anti-tumor effect of adoptively transferred NK cells was also improved. Thus sMIC could constitute a shed protein that could be targeted to enhance NK function.

The study is very interesting and demonstrates a variety of important points. Below please find some minor comments

1-Although the authors have presented convincingly the role of sMIC in inactivation of NK function, nevertheless it could have been exciting to demonstrate that sMIC+antibody complex does not have a direct activating effect on NK cells. These studies are very simple but could rule out any direct effect of the complex since many other complexes such as cytokine complex with antibodies have been hypothesized to be great activators of T and NK cells. Indeed it has been shown that IL-2/Ab complexes activate maturation and proliferation in CD8(+) T cells and natural killer (NK) cells to a much higher degree than conventional IL-2 therapy (Curr Pharm, Dec 2009;15(7):809-25. doi: 10.2174/138161209787582174).

2-Could the authors comment further on the differences they see in IFNγ gene induction between human and mice in regards to sMIC stimulation. This is a very important finding and it may lead to more of an understanding the dynamic differences between human and mice NK biology

3- TRAMP/MIC mice with high levels of tumor-shed sMIC developed poorly differentiated (PD) tumors whereas those with low levels of tumor-shed sMIC developed well-differentiated (WD) tumors. This observation is very exciting and could be due to differential growth rates of these tumors. Although gene signature for the cytotoxic and pro-inflammatory is clearly different in these two subsets of tumors, it could be quite interesting to understand the differences in IFN-γ induction.

4-The data especially the heatmap on polyfunctionality could be simplified more. Please explain the details of the figure, since it is quite difficult to understand the figure 6b.

5- Please provide the rationale for using NK-92 as the NK cells since these cells were shown to have decreased cytotoxicity when compared to NK cells. In addition, they lack secretion of IFN-γ.

6-Since the authors have found significant differences between well differentiated and poorly differentiated tumors in terms of the effect of sMIC, can perhaps the authors use these two different tumor models in their in vivo experiment to address the NK function? These studies will be a great adjunct to the manuscript or for the future experiments.

7-Although the paper is well written and presented there are a few omissions, for instance the sentence (Tumors often present with a highly heterogeneous) is omitted. Please make sure all the misspelling is corrected

Overall the paper is well written and well presented. The concept is novel and translational.

Reviewer #2:

Remarks to the Author:

This work deals with on the mechanism of tumor-derived NKG2D ligand soluble MIC (sMIC) in reprogramming NK cells with diminished cytotoxicity and polyfunctional potential, Also, these NK cells appear to secrete protumorigenic cytokines. Further, the use of an antibody targeting sMIC can influence the growth of xenogenic tumor in a murine model in conjunction with the use of NK-92 human lymphoma cell line.

This work aims to clarify the molecular mechanisms by which soluble NKG2DL can influence anti-tumor immune response of NK cells. It appears that NKG2DL can skew the features of NK cells, indeed, these cells appear not showing a cytotoxic and antitumor behavior. On the contrary, they can secrete cytokines that may have pro-tumorigenic properties. The analysis have been performed both in human and murine models.

This paper is an expansion of the findings described previously by the same research group and the large majority of relevant experiments to demonstrate the role of soluble NKG2DL MIC are performed with the anti-MIC antibody B10G5.

The main message of this paper is that the soluble MIC can activate the CBM-signalosome inflammatory pathways in NK cells while membrane MIC can activate NK cell cytotoxicity through activating PLC2y2/SLP-76/Vav1 pathway. This is shown mainly in figure 4 and 5. The experiments are performed in co-cultures with different kinds of C1R lymphoblastoid cell lines expressing mMICA or sMICA. As mentioned below, it is not clear whether NK cells and C1R cells are separated each other after the co-culture. In M and M section it is indicated that in some experiments have been used transwell and C1R cells or C1R-sMICA cells and in others cells were adherent through poly-L-Lysine.

It is obvious that the signal induced by a cell is different from that induced by a soluble molecule. Soluble molecule can be derived from shedding and exosomes production. In principle, I think that shedding inhibitors not always affect exosomes production. Thus, the presence of a membrane molecule can be found in the medium because it is released as an exosome-associated molecule. To state the different effect of sMICA and mMICA only, one should try to see whether the sMICA can induce a signal and the same can be done with an artificially linked mMICA at the plastic. Of course, this is not the ideal because cells can link to chemically reactive species present in the plastic but the complexity of interaction between two different cells is avoided.

The WB results are shown as cut and then assembled lanes. It is not clear whether the analysis of the different molecules have been performed upon stripping and re-probing of the same membrane with the different antibodies and whether the example of WB shown derive from the same experiment. Indeed, the density of beta actin used as control is really different in the different WB without a relative over-expression of the molecule analyzed, suggesting that the data shown derive from different experiments.

From previous works, it seems that the anti-MIC B10G5 can recognize MIC as either soluble or membrane expressed. Also, this antibody does not interfere with the NKG2D-mediated recognition of sMIC and this antibody can induce a slight ADCC effect with NK cells (the properties of this antibody are described in the results section of a previous paper and they are shown in a supplementary figure of the same papers).

The properties reported by the Authors are the following (extracted from their paper DOI: 10.1158/1078-0432.CCR-15-0845) : "Functional characterization of the anti-MIC mAb B10G5 The B10G5 mAb not only neutralizes free sMIC but also recognizes tumor cell surface membrane-bound MIC (Supplementary Fig. S1A and S1B), owing to sMIC sharing the same sequence and structure as the ectodomain of cell-surface MIC (16). Because B10G5 and the receptor NKG2D recognizes different epitopes of MIC (data not shown), B10G5 does not block the sensitivity of NKG2D-mediated NK cell cytolytic activity against MICp cells (Supplementary Fig. S1C). Conversely, B10G5 enhances the sensitivity of MICp-tumor cells to NK cell cytotoxicity (Supplementary Fig. S1C), presumably through antibody mediated cell cytotoxicity (ADCC) and/or enhanced immune synapse formation through simultaneously engaging NKG2D and Fc receptor on NK cells."

From the legend to figure S4 of the work <https://doi.org/10.1172/JCI69369>. "Data clearly demonstrate that B10G5 inhibits the binding of sMICB to NKG2D."

From these data extracted from two previous works, there is no a direct demonstration that NKG2D molecule is involved in the killing of TC2-MICB. Indeed, the cytotoxic activity of murine NK cells is not tested with an anti-NKG2D antibody. This should block the lysis of TC2-MICB if this lysis is only mediated by NKG2D-MICB interaction (but I think that several other molecules should be involved, of course). Further, cytolytic experiments using Fab'2 B10G5 antibody has not been performed to clarify the relevance of NKG2D-MICB interaction only, without the confounding effects due to Fcγ receptor engagement. Thus, the increase in cytolysis observed with the entire B10G5 mAb can be the result of both a positive signal (ADCC) due to interaction with the Fcγ receptor on NK cells and MICB on target cells and a negative signal covering with the antibody the

surface MICB on TC2-MICB cell line. Of course, the ADCC seems (as usual) stronger than the inhibiting signal and thus it is evident a slight increase of TC2-MICB sensitivity to NK cells. Also, it is not clearly stated whether TC2 and TC2-MICB cells show the same expression of several molecules involved in NK cell mediated recognition of target cells. This is essential to compare the bi-transgenic mice each other and to Tramp mice.

The mAb B10G5 is a murine antibody, I would think that some of the functional effects listed above can be observed only using murine NK cells and not human NK cells. For instance, the ADCC effect could be evident with murine NK cells expressing the Fcγ receptor (CD16), but not with human NK cells with low or no expression of CD16 such as NK-92. On the other hand, the NKL cell line express CD16 but of course it cannot use a murine antibody to kill human derived cell such as C1R. As a consequence of these considerations experiments performed with murine NK cells can have a contribution from the functional properties of B10G5 antibody, whereas the use of B10G5 in a human model would not interfere with the NKG2D-mediated recognition of either soluble or membrane MIC and this antibody should not trigger directly NK cell-mediated activity.

Based on these considerations the results shown both in a murine and in human models can represent effects differently regulated by the B10G5 antibody and dependent on different portion of the same antibody.

The analysis with sMIC both on murine and human NK cells has been done with activated NK cells. Indeed, murine NK cells have been activated with a huge amount of IL-2 (1000IU/ml) while Human NK cells have been stimulated and expanded with a not well defined Human Cell Expansion Kit. I am not used to culture murine NK cells and thus I can admit that this may be the usual amount of IL-2 used to activate NK cells. On the other hand, I am quite used to culture and expand human NK cells. The system used by the authors utilizes a kit which composition is not defined in the site of R and D system. Indeed, in the kit 5 NK cell expanders are used together with basal media. Of course these experiments can be repeated by everyone but it is essential to know with what the NK cells have been cultured. I understand that this kit appears to trigger NK cells expansion but it is not clear whether this expansion can determine the response observed to sMIC. Also, it appears that both murine and human NK cells have been stimulated with human sMIC. Murine NKG2D can interact with human MIC but of course this appears a quite artificial context to define the influence of an NKG2DL on murine NK cells.

It is not clear why the authors for these experiments have used activated NK cells. For human NK cells it is easy to get a lot of freshly isolated NK cells and use them directly. I suppose that is the same for murine NK cells isolated from the spleen, although I am not an expert on this point.

The experiment performed with double transgenic TRAMP/MIC mice with a low or high expression of sMIC are of interest. However, the physiological relevance of the findings found are difficult to be interpreted. Indeed, the expression of human MIC in murine cells is not only artificial but also it can interfere in an undefined manner with the interaction of murine natural ligands for NKG2D. This can alter the immune response of murine NK cells. The use of an antibody to block the effects observed can be dependent on the ability of the antibody to activate NK cells with the FC receptor. I did not find experiments in the previous papers published by the same research group using the Fab'2 of the anti-sMIC antibody. Further, one should be completely sure that the tumor cells obtained in the different transgenic mice express the main molecules involved in NK-tumor target interaction at the same level and in the same manner.

The same (the comparable expression of several molecules involved in NK-tumor target cell interaction) should be considered for the TRAMP-C2 (TC2) cells, sMICB-expressing TC2 cells (TC2-sMICB), and mMICB-expressing TC2 cells (TC2-mMICB) described in figure 3.

It is not clear whether after the co-cultures experiments between NK and target cells (for instance in figure 4) NK cells have been isolated and analyzed separately from the target cells. Also, the ratio between NK cells and target cells seems to be 1:5 for cytokine array (at least I understand that this specific ratio has been used: indeed from page 18 of the manuscript: "Primary mouse NK cells were co-cultured with mouse prostate tumor TC2 cell lines expressing soluble and membrane restricted MIC ligands, TC2-sMICB and TC2-mMICB respectively, at 1:5 ratios. Similarly, primary human NK cells were co-cultured with human B cell lymphoblast cell line C1R expressing soluble

and membrane restricted MIC ligands, C1R-sMICA and C1RmMICA respectively, at 1:5 ratios." While the ratio is 1:1 for the WB experiments. In any case the target and effectors should be separated before the analysis.

Minor points

Figure 1 panel a and b please insert the label as in panel c and d

Figure 2. It is not clear whether the only difference between soluble MIC_{low} and solubleMIC_{high} tumors is indeed the different expression of soluble MIC. Also, from previous work from the same group no analysis of several adhesion molecules and other molecule involved in the interaction with NK cells have been performed. Thus, the differences between the NK cell populations can be dependent on something other than the sMIC. The use of the antibody in this system would confuse the system because it can activate NK cells through the Fcγ receptor.

Figure 6. primary NK cells are resting or obtained with the system described above?

Figure 7 why not using primary freshly isolated (or activated) NK cells instead of NK-92 cells?

"For Lactate Dehydrogenase (LDH) cytotoxicity assay by a calorimetric quantification" calorimetric should be colorimetric, I think.

"Tumors ften present with" should be "tumors often present with", I think.

Reviewer #3:

Remarks to the Author:

MIC shedding is observed in a variety of tumors to evade NK and gamma-delta T cell surveillance via NKG2D. Inhibiting or clearing MIC shedding is a proven strategy to improve the cytotoxic efficacy of NK cells toward tumors. Using RNA sequencing analysis, the authors show a correlation between the exposure of soluble MIC (sMIC) and pro-inflammatory gene transcription. Furthermore, the authors demonstrate a reverse correlation between sMIC and membrane-bound MIC (mMIC) engagement using this technology. The specific mechanism of how these two forms of MIC regulate NK cell function remains poorly understood.

MIC shedding through cleavage by metalloproteases, resulting in the inhibition of NK cells is a well-known phenomenon. Furthermore, targeting sMIC using antibodies to improve anti-tumor responses has already been reported by this group (Lu et al., 2015; Wu, 2016; and Basher et al., 2020). The novelty of the current article is limited and is mainly related to the RNA profiling and the characterization of the different signaling pathways of the CBM signalosome complex involved in the NKG2D-sMIC interaction, and the PLCγ-VAV pathway regulated by the NKG2D-mMIC interaction.

Major comments

1) Comparison of the differentially regulated genes in NK cells following exposure to sMIC relative to mMIC is missing. In Figure 3, the authors show only limited cytokine profiling after 24 hours of incubation with either sMIC or mMIC. To better analyze the different molecular events of NK cells following sMIC vs. mMIC engagement, it would be better to compare short (~4h) and prolonged (~24h) incubation times following exposure to sMIC versus mMIC.

2) Chronic activation of NK cells due to a relatively long exposure of sMIC was mentioned by the authors, but was not compared to short exposure. Profiling the dysfunctional state of the NK cells following short vs. long exposure to sMIC will help to better reveal the molecular mechanism. This might include, for example, the switch from a CD45RA⁺CD45RO⁻ to a CD45RA⁻CD45RO⁺ phenotype and upregulation of inhibitory check points.

3) Figures 1a,b are not informative; out of dozens of genes upregulated or downregulated, the authors only describe a few selected genes (Fig. 1c,d). The authors should comment on the criteria for choosing these genes. Can a summary table of the 10-20 most highly regulated genes be provided? Are these selected genes among them?

4) In Figure 4, the authors used C1R cells (MIC-negative) to differentiate between sMIC and mMIC molecular signaling pathways. Expression of several of the signaling proteins presented was not significantly changed between the MIC negative relative to the mMIC expressing cells, and was even slightly higher in the MIC negative samples. Is it possible that this observation might be

explained by the relatively long NK;target co-incubation time of 30 min? Might a shorter co-incubation time change the protein/phosphorylation levels?

5) The proposed mechanism presented in Figure 7 should be described in the main text.

6) The proposed mechanism needs to be described in more detail, and substantiated with more experimental evidence.

ITAMs are known to activate downstream SLP76, PLC- γ , and VAV proteins. The recruitment and activation of PLC- γ , in turn, cleaves PIP2 into DAG and IP3. This mechanism provides the main signal for PKC θ activation. More experimental data are required to distinguish between the molecular signaling paths of short and prolonged sMIC or mMIC activation. Were other Src family kinases besides Fyn evaluated?

Minor comments

1) There was a significant increase in several genes related to cytotoxic processes, such as h-GzmM, h-CD160, h-NCR1, m-GzmB, m-IFN- γ , and m-CD160, following treatment of cells with sMIC+B10G5. What is the explanation for the increased activity? Were any Fc receptor blockers used?

A better understanding is required regarding the use of antagonistic antibody for NKG2D. Antagonistic antibody will block but not activate the NKG2D receptor, thus reversing the sMIC effect.

2) The serum concentration of sMIChi tumor (PD) (figure S3) ranges from 400 to 900 pg/ml, while the in vitro concentration of sMIC used in the RNAseq experiments was 100 ng/ml, which is several orders of magnitude higher. Can the authors explain why this concentration was chosen?

3) IFN- γ and TNF- α expression levels were not determined in any of the experiments. This information is essential since these are the most well known NK-derived cytokines.

4) The reference for purified recombinant sMICB (rsMICB) is mentioned for the first time on page 9. The authors need to clarify in which experiments recombinant sMIC was used.

5) In Figure 5, the authors used β -actin as the loading control, while evaluating signaling proteins involved in actin cytoskeleton reorganization. It would be preferable to use a different loading control that is not cytoskeleton related (e.g., GAPDH)?

Response to Critiques:

MS: COMMSBIO-20-3644-T

“Tumor-derived soluble NKG2D ligand sMIC reprograms NK cells to a dysfunctional inflammatory phenotype through activation of CBM signalosome” authored by Payal Dhar, Fahmin Basher, Zhe Ji, Lei Huang, Si Qin, Derek A. Wainwright, Jerid Robinson, Shaye Hagler, Jing Zhou, Sean MacKay, Jennifer D. Wu

We thank the reviewers for your recognition of the significance of our study and your critical reading of the manuscript. We have now addressed all your comments point by point below. All the revisions are highlighted **YELLOW** in the revised manuscript.

Response to Critique of Reviewer #1:

In this manuscript Dhar et al presented evidence for the role of sMIC in inactivation of NK cell function. They presented evidence that sMIC reprogramed NK cell to secrete pro-tumorigenic cytokines with diminished cytotoxicity and decreased polyfunctionality. They further demonstrated that antibody clearing sMIC restored the NK cell function. sMIC was also found to selectively activate the CBM-signalosome inflammatory pathways in NK cells. In addition, tumor cell membrane-bound MIC (mMIC) stimulated NK cell function through PLC2 γ 2/SLP-76/Vav1 pathway. In vivo anti-tumor effect of adoptively transferred NK cells was also improved. Thus sMIC could constitute a shed protein that could be targeted to enhance NK function.

The study is very interesting and demonstrates a variety of important points. Below please find some minor comments

Critique 1: Although the authors have presented convincingly the role of sMIC in inactivation of NK function, nevertheless it could have been exciting to demonstrate that sMIC+antibody complex does not have a direct activating effect on NK cells. These studies are very simple but could rule out any direct effect of the complex since many other complexes such as cytokine complex with antibodies have been hypothesized to be great activators of T and NK cells. Indeed it has been shown that IL-2/Ab complexes activate maturation and proliferation in CD8(+) T cells and natural killer (NK) cells to a much higher degree than conventional IL-2 therapy (Curr Pharm, Dec 2009;15(7):809-25. doi: 10.2174/138161209787582174).

Response:

Based on our current findings, sMIC+B10G5 complex alone does not have a direct effect on NK cell activation. As demonstrated in the data shown below, no significant differences were observed in IFN γ production and granzyme B expression in human NK cells stimulated with recombinant soluble MIC (sMIC) and the sMIC+ B10G5 complex, confirming that the complex by itself does not activate NK cells.

Critique 2: Could the authors comment further on the differences they see in IFN γ gene induction between human and mice in regards to sMIC stimulation. This is a very important finding and it may lead to more of an understanding the dynamic differences between human and mice NK biology

Response:

Thank you for bringing up this point. We have evaluated the IFN γ production in primary human and mouse NK cells in response to sMIC and mMIC stimulation using the co-culture experimental system at various time points. At all time points being investigated, both human and mouse NK cells produce significantly more IFN γ when stimulated by mMIC than being stimulated by MIC-negative tumor cells or tumor cells expressing sMIC (Supplement data **Figure S7a-c**). Please note, although NK cell showed an interesting dynamics in IFN γ production as presented, as this is an in vitro experimental in a given setting, we are reluctant to make a conclusion of the dynamics related to in vivo NK cell function over the observation.

Critique 3: TRAMP/MIC mice with high levels of tumor-shed sMIC developed poorly differentiated (PD) tumors whereas those with low levels of tumor-shed sMIC developed well-differentiated (WD) tumors. This observation is very exciting and could be due to differential

growth rates of these tumors. Although gene signature for the cytotoxic and pro-inflammatory is clearly different in these two subsets of tumors, it could be quite interesting to understand the differences in IFN-g induction.

Critique 6: Since the authors have found significant differences between well differentiated and poorly differentiated tumors in terms of the effect of sMIC, can perhaps the authors use these two different tumor models in their in vivo experiment to address the NK function? These studies will be a great adjunct to the manuscript or for the future experiments.

Responses to Critiques 3 and 6:

- a) Thank you for the comments and great insights. As we have described in our published studies characterizing the TRAMP/MICB mouse model [1], the poorly differentiated tumors (PD) grew significantly more aggressive than the well-differentiated tumors. We also described that the PD tumors have lost most of the surface MIC from IHC analyses and was associated with elevated tumor interstitial and serum sMIC. Thus, our interpretation was that tumor shedding sMIC was the most significant contributor to the increased serum level of sMIC rather than simply tumor volume, as MIC is mostly localized to the tumor cell surface in WD tumors (Fig 2a of Liu et al, JCI 2013, ref #6).
- b) We have included the flow cytometry data (**Figure 2h**) in the revision demonstrating that TIL-NK cells from sMIC^{hi} tumors presented diminished IFN γ production and other effector molecules (TNF α and Granzyme B).

Critique 4: The data especially the heatmap on polyfunctionality could be simplified more. Please explain the details of the figure, since it is quite difficult to understand the figure 6b.

Response:

The details of the single cell polyfunctionality heatmap have been modified in the figure legend (Fig 6b). The format of the single cell polyfunctionality heatmap and the explanation is based on the standard methods as published in a number of high impact publications represented in the attached references [2, 3].

Critique 5: Please provide the rationale for using NK-92 as the NK cells since these cells were shown to have decreased cytotoxicity when compared to NK cells. In addition, they lack secretion of IFN-g.

Response:

We agree with your view of the limitations of NK -92 cells. We used NK-92 cells in this study solely as a “tool” cell line for the purpose of proof-of-concept, to demonstrate that mMIC stimulates NK cell cytotoxicity and that sMIC negatively impacts NK cell cytotoxicity.

Critique 7: Although the paper is well written and presented there are a few omissions, for instance the sentence (Tumors ften present with a highly heterogenous) o is omitted. Please make sure all the misspelling is corrected.

Response:

We apologize for the typos. We have made corrections throughout the manuscript. Thank you for your careful review.

Overall the paper is well written and well presented. The concept is novel and translational.

Response: We again thank the reviewer for their positive comments and for recognizing the significance of our study.

Response to Critique of Reviewer #2:

Major Critique 1: The main message of this paper is that the soluble MIC can activate the CBM-signalosome inflammatory pathways in NK cells while membrane MIC can activate NK cell cytotoxicity through activating PLC2 γ 2/SLP-76/Vav1 pathway. This is shown mainly in figure 4 and 5. The experiments are performed in co-cultures with different kinds of C1R lymphoblastoid cell lines expressing mMICA or sMICA. As mentioned below, it is not clear whether NK cells and C1R cells are separated each other after the co-culture. In M and M section it is indicated that in some experiments have been used transwell and C1R cells or C1R-sMICA cells and in others cells were adherent through poly-L-Lysine.

Response:

- a) We appreciate your concern. To circumvent the potential tumor cell contamination in NK cell collection for downstream biochemical/signaling assays, we coated the plates with poly-L-Lysine to ensure that tumor cells were truly adherent to the plates and that the collected suspension cells only contained NK cells. We confirmed that there is no tumor cell contamination in collected NK cells by re-probing the western-blot with the anti-GFP antibody, since all tumor cells were expressing GFP as a reporter and sorting marker for mMICA or sMICA overexpression. No GFP signal was shown on the blots we have analyzed. We have now included these data as the **Supplemental Figure S5b and S5c** in the revision.
- b) We apologize for the unclarity. We have edited and incorporated additional methodology details in the M&M section.

Major Critique 2: It is obvious that the signal induced by a cell is different from that induced by a soluble molecule. Soluble molecule can be derived from shedding and exosomes production. In principle, I think that shedding inhibitors not always affect exosomes production. Thus, the presence of a membrane molecule can be found in the medium because it is released as an exosome-associated molecule. To state the different effect of sMICA and mMICA only, one should try to see whether the sMICA can induce a signal and the same can be done with an artificially linked mMICA at the plastic. Of course, this is not the ideal because cells can link to chemically reactive species present in the plastic but the complexity of interaction between two different cells is avoided.

Response:

We have now included the western blot analyses of the key representative signaling molecules of cytotoxicity and cytokine pathways using the experimental setting where primary human NK cells were stimulated with recombinant sMIC and plate-bound MIC upon various time points of stimulation (**Fig S5d and S6e**).

Phosphorylation of the NF- κ B subunit p65 was evidently elevated in NK cells stimulated with recombinant sMIC as compared to stimulation with plate-bound MIC at 60 minutes of stimulation time period. Differences in ADAP levels were also evidently higher in recombinant sMIC-stimulated NK cells compared to plate-bound MIC with 10 minutes and 30 minutes of stimulation. These data are consistent with the concept that sMIC stimulation preferably activates NK cell pro-inflammatory pathways. A higher level of JNK and ERK was seen with plate-bound MIC than sMIC stimulation at short period of time (10 mins). However, at 60 mins of stimulation, phosphorylation of JNK and ERK reached a similar level with sMIC and plate-bound MIC stimulation. As kinase phosphorylation is often in a dynamic recycling state and chemical cross-link on the plates could also have an impact, we are reluctant to make a conclusive statement of the findings.

Major Critique 3: The WB results are shown as cut and then assembled lanes. It is not clear whether the analysis of the different molecules have been performed upon stripping and re-probing of the same membrane with the different antibodies and whether the example of WB shown derive from the same experiment. Indeed, the density of beta actin used as control is really different in the different WB without a relative over-expression of the molecule analyzed, suggesting that the data shown derive from different experiments.

Response:

Individual phospho-proteins, their respective total proteins and beta- actin were presented from the same blots. In details, the blots were first probed for phospho-proteins, followed by stripping and re-probing for respective total proteins and subsequently for b-actin.

Since the molecular weights of some of the molecules are very similar, for example, pFyn (60kDa), phospho NF- κ B (65kDa); ADAP (130kDa), CARD11 (130kDa), MALT1 (90kDa), pPKC θ (79kDa), samples from the same experiment were divided for running on different gels/blots. These details have been incorporated in the figure legends of main **Figure 4** and **Figure 5**.

Major Critique 4: From these data extracted from two previous works, there is no a direct demonstration that NKG2D molecule is involved in the killing of TC2-MICB. Indeed, the cytotoxic activity of murine NK cells is not tested with an anti-NKG2D antibody. This should block the lysis of TC2-MICB if this lysis is only mediated by NKG2D-MICB interaction (but I think that several other molecules should be involved, of course).

Response:

As demonstrated in new supplemental figures, **Figures S6b**, NK cells from NKG2D-deficient mice (NKG2D $^{-/-}$) did not present a difference in cytotoxicity against TC2-mMICB cells and TC2 cells, suggesting that the high cytotoxic activity of NK cells against TC2-mMICB cells is mediated through NKG2D. Furthermore, as shown in **Fig S7c**, the mMIC stimulation- mediated enhanced

IFN γ production by NK cells was also abrogated upon NKG2D blocking, further confirming a NKG2D-dependent effect.

Major Critique 5: Further, cytolytic experiments using Fab'2 B10G5 antibody has not been performed to clarify the relevance of NKG2D-MICB interaction only, without the confounding effects due to Fc γ receptor engagement. Thus, the increase in cytolysis observed with the entire B10G5 mAb can be the result of both a positive signal (ADCC) due to interaction with the Fc γ receptor on NK cells and MICB on target cells and a negative signal covering with the antibody the surface MICB on TC2-MICB cell line. Of course, the ADCC seems (as usual) stronger than the inhibiting signal and thus it is evident a slight increase of TC2-MICB sensitivity to NK cells.

Major Critique 7: The mAb B10G5 is a murine antibody, I would think that some of the functional effects listed above can be observed only using murine NK cells and not human NK cells. For instance, the ADCC effect could be evident with murine NK cells expressing the Fc γ receptor (CD16), but not with human NK cells with low or no expression of CD16 such as NK-92. On the other hand, the NKL cell line express CD16 but of course it cannot use a murine antibody to kill human derived cell such as C1R. As a consequence of these considerations experiments performed with murine NK cells can have a contribution from the functional properties of B10G5 antibody, whereas the use of B10G5 in a human model would not interfere with the NKG2D-mediated recognition of either soluble or membrane MIC and this antibody should not trigger directly NK cell-mediated activity.

Based on these considerations the results shown both in a murine and in human models can represent effects differently regulated by the B10G5 antibody and dependent on different portion of the same antibody.

Response to Major Critiques 5 and 7:

Thank you for the discussion. To clarify, B10G5 is a murine IgG1 which is known to confer minimal ADCC effect for mouse NK cells and human NK cells. Only murine IgG2a and IgG3 would present significant cross-reactivity with mouse or human FcR. Moreover, NK-92 cells lack FcR (CD16) and do not confer ADCC function.

Major Critique 6: Also, it is not clearly stated whether TC2 and TC2-MICB cells show the same expression of several molecules involved in NK cell mediated recognition of target cells. This is essential to compare the bi-transgenic mice each other and to Tramp mice.

Response:

We analyzed the expression of the other major mouse NKG2D ligands on TC2, TC2-sMICB and TC2-mMICB cells by flow cytometry. No significant differences in the expression level were observed among these cell lines. We have now included this data in the Supplemental section (**Figure S4b**) in the revision. These data suggest that different forms of MIC expression (mMIC vs. sMIC) are the drivers of differential impact on NK cell function.

Major Critique 8: The analysis with sMIC both on murine and human NK cells has been done with activated NK cells. Indeed, murine NK cells have been activated with a huge amount of IL-2 (1000IU/ml) while Human NK cells have been stimulated and expanded with a not well

defined Human Cell Expansion Kit. I am not used to culture murine NK cells and thus I can admit that this may be the usual amount of IL-2 used to activate NK cells. On the other hand, I am quite used to culture and expand human NK cells. The system used by the authors utilizes a kit which composition is not defined in the site of R and D system. Indeed, in the kit 5 NK cell expanders are used together with basal media. Of course these experiments can be repeated by everyone but it is essential to know with what the NK cells have been cultured. I understand that this kit appears to trigger NK cells expansion but it is not clear whether this expansion can determine the response observed to sMIC. Also, it appears that both murine and human NK cells have been stimulated with human sMIC. Murine NKG2D can interact with human MIC but of course this appears a quite artificial context to define the influence of an NKG2DL on murine NK cells.

It is not clear why the authors for these experiments have used activated NK cells. For human NK cells it is easy to get a lot of freshly isolated NK cells and use them directly. I suppose that is the same for murine NK cells isolated from the spleen, although I am not an expert on this point.

Response:

- (a) The use of IL-2 (1000IU/ml) expansion of mouse NK cells is a well-established and accepted approach described in literature [4]. Resting naïve murine NK cells freshly isolated from mice is commonly known to present no or nominal cytotoxicity in killing target cells. The use of in vitro activated NK cells for cytotoxic function assay is well accepted in the literature.
- (b) For expansion of human NK cells, due to proprietary concerns, the vendor (R&D) cannot disclose the composition of the human NK cell expanders. However, as an example, the use of CellXVivo Human B cell Expansion kit has been peer-reviewed and published in literature [5, 6].
- (c) Due to limited access to large amount of fresh blood samples at need for research in our facility setting, we chose to obtain cells commercially available frozen PBMCs. These frozen cells need to be revived and expanded before experimental use. Using ex vivo expanded human NK cells for use in research or even clinical settings is a common practice described in literature.

Major Critique 9: The experiment performed with double transgenic TRAMP/MIC mice with a low or high expression of sMIC are of interest. However, the physiological relevance of the findings found are difficult to be interpreted. Indeed, the expression of human MIC in murine cells is not only artificial but also it can interfere in an undefined manner with the interaction of murine natural ligands for NKG2D. This can alter the immune response of murine NK cells. The use of an antibody to block the effects observed can be dependent on the ability of the antibody to activate NK cells with the FC receptor. I did not find experiments in the previous papers published by the same research group using the fab'2 of the anti-sMIC antibody. Further, one should be completely sure that the tumor cells obtained in the different transgenic mice express the main molecules involved in NK-tumor target interaction at the same level and in the same manner.

Response:

- (a) All animal models, whether genetically engineered, humanized, or transplanted tumor models, are all artificial models being used to address relevant biological questions. Using mouse models to study the principle of a biology is a commonly accepted practice. One may consider that genetically engineered mouse model is artificial, not as nature as human, but this is as close as possible to address the underlying science in an immune competent host other than using human as an ultimate experimental model.
- (b) The biology presented in TRAMP/MIC mouse model had been peer-reviewed and published in J. Clinical Investigation [1]. A number of subsequent peer-reviewed studies with this model have been published in well represented Journals, such as Clinical Cancer Research (ref 30), Zhang et al., 2018, Journal for ImmunoTherapy of Cancer [7], and Zhang et al, 2017, Science Advances.
- (c) The relevance of MIC/NKG2D axis and tumor shedding sMIC in human cancer patients were published as early as 2002 (Groh et.al, 2002, Nature). Since then, a vast amount of correlative studies were published demonstrating the correlation of sMIC with CD8 T, NK, NKT, gamma-delta T cells and clinical prognosis in cancer patients. However, these correlative studies only presented a “correlation” or a “relevance”. Due to the fact that mice lack MIC gene and that murine NKG2D ligands were structurally different and regulated differently from human MIC [8], there had not been any progress on understanding how human sMIC impacts tumor immunity and no means to validate whether sMIC is a therapeutic target before the creation of the TRAMP/MIC mouse model. To our view, the creation of the TRAMP/MIC mouse model has offered an invaluable tool to understand the biology of sMIC/MIC during tumor progression and a pre-clinical model for evaluating therapies targeting the MIC/NKG2D pathway.
- (d) To emphasize again, B10G5 is a murine IgG1. Its ability to mediate ADCC is minimal. The mechanism of action of B10G5 were described in a number of publications (Lu et al, CCR, 2015; Zhang et al, Science advances, 2017; Zhang et al., JITC; 2018).

Major Critique 10: The same (the comparable expression of several molecules involved in NK-tumor target cell interaction) should be considered for the TRAMP-C2 (TC2) cells, sMICB-expressing TC2 cells (TC2-sMICB), and mMICB-expressing TC2 cells (TC2-mMICB) described in figure 3

Response: Please refer to the responses to Major Critique 6.

Major Critique 11: It is not clear whether after the co-cultures experiments between NK and target cells (for instance in figure 4) NK cells have been isolated and analyzed separately from the target cells. Also, the ratio between NK cells and target cells seems to be 1:5 for cytokine array (at least I understand that this specific ratio has been used: indeed from page 18 of the manuscript:” Primary mouse NK cells were co-cultured with mouse prostate tumor TC2 cell lines expressing soluble and membrane restricted MIC ligands, TC2-sMICB and TC2-mMICB respectively, at 1:5 ratios. Similarly, primary human NK cells were co-cultured with human B cell lymphoblast cell line C1R expressing soluble and membrane restricted MIC ligands, C1R-sMICA and C1RmMICA

respectively, at 1:5 ratios.” While the ratio is 1:1 for the WB experiments. In any case the target and effectors should be separated before the analysis.

Response:

The cytokine array experiments had been performed using target and effector cell ratios of 1:1 and 1:5. The results demonstrated similar trends in the cytokine profiles at the two E:T ratios. Please see below representative data from E:T 1:1. The same conclusion can be drawn as E:T ratio 1:5. We thus only representatively showed 1:5 E:T.

Minor Critique 1: Figure 1 panel a and b please insert the label as in panel c and d

Response:

For Figure 1 panel a and b, same labels cannot be inserted because Fig 1 panel a and b represent compiled overview of thousands of differentially expressed genes in human and mouse NK cells with different stimulation conditions, whereas panel c and d highlight the main representative differentially expressed genes associated with cytotoxicity and pro-inflammatory functions (as labelled).

Minor Critique 2: Figure 2. It is not clear whether the only difference between soluble MIClow and solubleMIChigh tumors is indeed the different expression of soluble MIC. Also, from previous work from the same group no analysis of several adhesion molecules and other molecule involved in the interaction with NK cells have been performed. Thus, the differences between the NK cell populations can be dependent on something other than the sMIC. The use of the antibody in this system would confuse the system because it can activate NK cells through the Fcγ receptor.

Response:

Yes, it is true that biology is a multi-dimensional complex, particularly with in vivo systems and in humans. In this sense, the transgenic mouse model in which many genetic factors are controlled within a colony is much more conducive to address the potential impact of a single factor. It is no doubt or a given that many factors in the biological system can impact NK cell function. However, the focus of this current investigation is to address the impact of sMIC on NK cell function in a more in-depth molecular level. The in vivo findings were validated rigorously and well supported by in vitro assays in this study.

Minor Critique 3: Figure 6. primary NK cells are resting or obtained with the system described above?

Response:

Primary NK cells represented in Figure 6 were obtained with the same system as described above

Minor Critique 4: Figure 7 why not using primary freshly isolated (or activated) NK cells instead of NK-92 cells?

Response:

NK-92 cells were used in this proof-of-concept study in that: 1) they do not confer ADCC effect due to lack of Fc; 2) are readily available and easy to expand.

Minor Critique 5: “For Lactate Dehydrogenase (LDH) cytotoxicity assay by a calorimetric quantification” calorimetric should be colorimetric, I think.

Response:

We have corrected the typo. Thank you for your careful review.

Minor Critique 6: “Tumors ften present with” should be “tumors often present with”, I think.

Response:

We have corrected the typo. Thank you for your careful review.

Response to Critique of Reviewer #3:

MIC shedding is observed in a variety of tumors to evade NK and gamma-delta T cell surveillance via NKG2D. Inhibiting or clearing MIC shedding is a proven strategy to improve the cytotoxic efficacy of NK cells toward tumors. Using RNA sequencing analysis, the authors show a correlation between the exposure of soluble MIC (sMIC) and pro-inflammatory gene transcription. Furthermore, the authors demonstrate a reverse correlation between sMIC and membrane-bound MIC (mMIC) engagement using this technology. The specific mechanism of how these two forms of MIC regulate NK cell function remains poorly understood.

MIC shedding through cleavage by metalloproteases, resulting in the inhibition of NK cells is a well-known phenomenon. Furthermore, targeting sMIC using antibodies to improve anti-tumor responses has already been reported by this group (Lu et al., 2015; Wu, 2016; and Basher et al., 2020). The novelty of the current article is limited and is mainly related to the RNA profiling and the characterization of the different signaling pathways of the CBM signalosome complex involved in the NKG2D–sMIC interaction, and the PLC γ -VAV pathway regulated by the NKG2D-mMIC interaction.

Major Critique 1: Comparison of the differentially regulated genes in NK cells following exposure to sMIC relative to mMIC is missing. In Figure 3, the authors show only limited cytokine profiling after 24 hours of incubation with either sMIC or mMIC. To better analyze the different molecular events of NK cells following sMIC vs. mMIC engagement, it would be better to compare short (~4h) and prolonged (~24h) incubation times following exposure to sMIC versus mMIC.

Response:

- (a) The RNA sequencing analysis was initially performed to address the impact of sMIC on NK cells, with the presence and absence of sMIC clearing antibody for validation. We agree that, to address the differential impact of sMIC and mMIC on NK cell reprogramming, ideally one would use the co-culture system of NK cells co-cultured with tumor cells expressing sMIC and tumor cells expressing mMIC respectively. Scientifically, this is rather challenging to ensure that there will not be any contamination of tumor RNA for analyses. We only included experiments with recombinant sMIC in the presence or absence of the sMIC clearing antibody as a starting point to gain a landscape of the gene expression with or without sMIC. We subsequently addressed the differential impact of sMIC and mMIC on NK cell function different experiments whenever technically feasible. To our view, the lack of mMIC stimulation RNAseq data does not compromise the study findings.
- (b) We appreciate reviewer's comment on the dynamics of the cytokine production and gene profiling. Using qRT-PCR analyses of key function-related genes in human NK cells stimulated with sMIC and plate-bound MIC for 6h and 18h durations, a time-dependent dynamic in the expression of Klrk1 (NKG2D) and inflammatory CCL3 was shown below. NKG2D expression was significantly downregulated with prolonged sMIC stimulation compared to plate-bound MIC stimulation. CCL3 gene expression was consistently significantly upregulated with sMIC stimulation compared to plate-bound MIC stimulation. From our data presented in **Supplement Figures S5d and S6e**, one may consider that plate-bound MIC stimulation mimics mMIC stimulation to some extent, although it is impossible to assure to what extent, as effector-target cell interaction is much more complex than the simple MIC/NKG2D interaction.

We have also included key data from cytokine array to demonstrate production of different cytokines could be time dependent in Supplemental **Figure S4a**. Regardless, sMIC stimulation consistently resulted in markedly high production of pro-inflammatory cytokines/chemokines although the absolute quantity varies overtime.

** $p < 0.01$, *** $p < 0.001$ and represent significant differences between NK+sMIC and NK+plate-bound MIC at respective time points.

Major Critique 2: Chronic activation of NK cells due to a relatively long exposure of sMIC was mentioned by the authors, but was not compared to short exposure. Profiling the dysfunctional state of the NK cells following short vs. long exposure to sMIC will help to better reveal the molecular mechanism. This might include, for example, the switch from a CD45RA⁺CD45RO⁻ to a CD45RA⁻CD45RO⁺ phenotype and upregulation of inhibitory check points.

Response:

We evaluated the surface expression of the major NK cell inhibitory and activating receptors as well as markers of NK maturation at early (1 hour) and late (36 hours) time-points in co-culture experimental system. We did not observe any significant differences in the markers at either time point. The data are shown below.

Major Critique 3: Figures 1a,b are not informative; out of dozens of genes upregulated or downregulated, the authors only describe a few selected genes (Fig. 1c,d). The authors should comment on the criteria for choosing these genes. Can a summary table of the 10-20 most highly regulated genes be provided? Are these selected genes among them?

Response:

There are 6,503 genes differentially expressed in sMIC-stimulated versus unstimulated mouse NK cells. Gene ontology analysis has revealed significantly differentially expressed genes were associated with mitochondrial metabolism, autophagy, NK cell survival genes, and NK functional genes. Genes related to NK cell metabolic function, survival (maintenance) are currently under active validation and investigation in the lab. As such, we do not feel that we have sufficient evidence to list those top metabolic/autophagy related genes until they are validated and better understood in our ongoing and future studies. The raw data reading of these genes is included in the submitted RAW DATA excel file.

The genes highlighted in the heatmaps are the significant differentially expressed essential genes related to NK cell function identified by gene ontology analysis. Criteria of selection of these key represented NK cell functional genes were based on fold change cutoff of 2 and FDR-adjusted p-value less than 0.05.

Major Critique 4: In Figure 4, the authors used C1R cells (MIC-negative) to differentiate between sMIC and mMIC molecular signaling pathways. Expression of several of the signaling proteins presented was not significantly changed between the MIC negative relative to the mMIC expressing cells, and was even slightly higher in the MIC negative samples. Is it possible that this observation might be explained by the relatively long NK:target co-incubation time of 30 min? Might a shorter co-incubation time change the protein/phosphorylation levels?

Response:

Thank you for the comments. We have performed experiments at 10 mins vs. 30 mins stimulation. As shown below, the differences in phosphorylation of the signaling molecules with sMIC and mMIC stimulation at 10 min were not as pronounced at 30 mins. Theoretically, this is expected, as the stimulation with sMIC would be more chronic, whereas the stimulation of mMIC would be more acute interaction directed by the formation of immune synapse between NK cells and target cells. As we know, the signaling involving in immune synapse formation is a complicated process, often involves recycling of molecules and thus a timing dependent phosphorylation of certain signaling mediators.

Major Critique 5: The proposed mechanism presented in Figure 7 should be described in the main text.

Response: The description of the proposed mechanism is now included in the main text Fig 8.

Major Critique 6: The proposed mechanism needs to be described in more detail, and substantiated with more experimental evidence. ITAMs are known to activate downstream SLP76, PLC- γ , and VAV proteins. The recruitment and activation of PLC- γ , in turn, cleaves PIP2 into DAG and IP3. This mechanism provides the main signal for PKC θ activation. More experimental data are required to distinguish between the molecular signaling paths of short and prolonged sMIC or mMIC activation. Were other Src family kinases besides Fyn evaluated?

Response:

Thank you for the suggestions. We have evaluated activation of signaling molecules DAG lipase β (there is no DAG-specific antibody available commercially), phospho-IP3, and Src kinase Lck in human NK cells at 10 minutes and 30 minutes sMIC and mMIC stimulation time points (revised main **Figure 4f** and supplementary **Fig S5e**). There is no apparent difference in Lck activation levels with sMIC versus mMIC at both time points, suggesting the possibility that Lck activation might be associated with both cytotoxicity and cytokine signaling events. However, DAG lipase β presented a dynamic with different stimulation. At 10 minutes, DAG was comparable between sMIC and mMIC stimulation. At 30 minutes, DAG was evidently higher upon sMIC stimulation. As DAG is the upstream molecule of CBM signalosome pathway, upstream of PKC- θ , the observation is consistent with the concept being described here that chronic sMIC stimulation preferably activates NK cell pro-inflammatory pathways. On the contrary, phosphorylation of IP3 was markedly elevated in NK cells stimulated with mMIC with shorter stimulation period. It is understood that phosphorylation of IP3 is related to Ca⁺⁺ influx to initiate NK cell cytotoxic function and that this event reaches its pinnacle with 5 mins [9, 10]. Thus, the observed dynamic of phospho-IP3 is anticipated.

Minor Critique 1: There was a significant increase in several genes related to cytotoxic processes, such as h-GzmM, h-CD160, h-NCR1, m-GzmB, m-IFN- γ , and m-CD160, following treatment of cells with sMIC+B10G5. What is the explanation for the increased activity? Were any Fc receptor blockers used? A better understanding is required regarding the use of antagonistic antibody for NKG2D. Antagonistic antibody will block but not activate the NKG2D receptor, thus reversing the sMIC effect.

Response:

- a. We have now included key cytotoxicity assay and western blot data in the supplemental section (**Supplemental Figure S5a and S6a**), which clearly indicate and confirm our previous findings of NKG2D-dependent NK cell cytotoxicity functions and signaling pathways. Pre-incubation with the NKG2D-specific blocking antibody 1D11 diminished the cytotoxicity of NK cells against C1R-mMICA cells (**Supplemental Fig S6a**). The differences in the activation of key signaling molecules of cytokine pathways upon sMIC and mMIC stimulation were also abrogated (**Supplemental Fig S5a**).
- b. Please note that there are no significant differences in the gene expression of the cytotoxicity associated genes in NK cells stimulated with sMIC+B10G5 compared to

unstimulated NK cells, suggesting that B10G5 rescues the NK cell function to the normal state in the presence of sMIC.

Minor Critique 2: The serum concentration of sMIC_{hi} tumor (PD) (figure S3) ranges from 400 to 900 pg/ml, while the in vitro concentration of sMIC used in the RNAseq experiments was 100 ng/ml, which is several orders of magnitude higher. Can the authors explain why this concentration was chosen?

Response:

Please note that the sMIC concentration in the tumors is anticipated to be significantly much higher (at least 100 fold higher) compared to that in the serum. The sMIC concentration we were using (100 ng/ml) in in vitro assay is anticipated to be similar to that in the tumors.

Minor Critique 3: IFN- γ and TNF- α expression levels were not determined in any of the experiments. This information is essential since these are the most well-known NK-derived cytokines.

Response: Please refer to our detailed response to **Reviewer 1 Critique 2, 3 and 6**. Briefly, we have included flow cytometry data of sMIC-*hi* and sMIC-*lo* tumor infiltrating NK cells and demonstrated that NK cells in sMIC-*hi* tumors presented impaired ability to produce IFN γ and TNF α (Revised **Figure 2h**).

Minor Critique 4: The reference for purified recombinant sMICB (rsMICB) is mentioned for the first time on page 9. The authors need to clarify in which experiments recombinant sMIC was used.

Response: We apologize for the unclarity. The details have now been highlighted in the main text.

Minor Critique 5: In Figure 5, the authors used β -actin as the loading control, while evaluating signaling proteins involved in actin cytoskeleton reorganization. It would be preferable to use a different loading control that is not cytoskeleton related (e.g., GAPDH)?

Response: We re-probed the blots for representative signaling molecules with GAPDH, no differences were observed. Please find the data below.

References

1. Liu, G., et al., *Perturbation of NK cell peripheral homeostasis accelerates prostate carcinoma metastasis*. J Clin Invest, 2013. **123**(10): p. 4410-22.
2. Zhu, H., et al., *Metabolic Reprograming via Deletion of CISH in Human iPSC-Derived NK Cells Promotes In Vivo Persistence and Enhances Anti-tumor Activity*. Cell Stem Cell, 2020. **27**(2): p. 224-237 e6.
3. Parisi, G., et al., *Persistence of adoptively transferred T cells with a kinetically engineered IL-2 receptor agonist*. Nat Commun, 2020. **11**(1): p. 660.
4. Regunathan, J., et al., *Differential and nonredundant roles of phospholipase Cgamma2 and phospholipase Cgamma1 in the terminal maturation of NK cells*. J Immunol, 2006. **177**(8): p. 5365-76.
5. Yi, G., et al., *A DNA Vaccine Protects Human Immune Cells against Zika Virus Infection in Humanized Mice*. EBioMedicine, 2017. **25**: p. 87-94.
6. Weagel, E.G., et al., *Biomarker analysis and clinical relevance of TK1 on the cell membrane of Burkitt's lymphoma and acute lymphoblastic leukemia*. Onco Targets Ther, 2017. **10**: p. 4355-4367.
7. Zhang, J., et al., *Antibody targeting tumor-derived soluble NKG2D ligand sMIC provides dual co-stimulation of CD8 T cells and enables sMIC(+) tumors respond to PD1/PD-L1 blockade therapy*. J Immunother Cancer, 2019. **7**(1): p. 223.
8. Guerra, N., et al., *NKG2D-deficient mice are defective in tumor surveillance in models of spontaneous malignancy*. Immunity, 2008. **28**(4): p. 571-80.
9. Maul-Pavicic, A., et al., *ORAI1-mediated calcium influx is required for human cytotoxic lymphocyte degranulation and target cell lysis*. Proc Natl Acad Sci U S A, 2011. **108**(8): p. 3324-9.
10. Jamieson, A.M., et al., *The role of the NKG2D immunoreceptor in immune cell activation and natural killing*. Immunity, 2002. **17**(1): p. 19-29.

Reviewers' Comments:

Reviewer #1:

Remarks to the Author:

The authors have answered my comments and questions satisfactorily and revised the manuscript accordingly, and therefore I recommend publication of the paper.

thank you the opportunity to review the paper

Reviewer #2:

Remarks to the Author:

Dear Editor, the authors have replied to my concerns enough well. The direct demonstration that the different transgenic mice can indeed express at the same level and with similar function several other molecules involved in NK cell activity is not demonstrated, in my opinion; I do not have any other specific comment on this manuscript.

I do not have any specific negative evaluation of this manuscript.

I am not against the endorsement for publication of this manuscript

Reviewer #3:

Remarks to the Author:

All questions have been adequately addressed in the revised manuscript.